# World-To-Image: Grounding Text-to-Image Generation with Agent-Driven World Knowledge

## Abstract

While text-to-image (T2I) models can synthesize high-quality images, their performance degrades significantly when prompted with novel or out-of-distribution (OOD) entities due to inherent knowledge cutoffs. We introduce World-To-Image, a novel framework that bridges this gap by empowering T2I generation with agent-driven world knowledge. We design an agent that dynamically searches the web to retrieve images for concepts unknown to the base model. This information is then used to perform multimodal prompt optimization, steering powerful generative backbones toward an accurate synthesis. Critically, our evaluation goes beyond traditional metrics, utilizing modern assessments like LLM-Grader and ImageReward to measure true semantic fidelity. Our experiments show that World-To-Image substantially outperforms state-of-the-art methods in both semantic alignment and visual aesthetics, achieving **+8.1%** improvement in accuracy-to-prompt on our curated NICE benchmark. Our framework achieves these results with high efficiency in less than three iterations, paving the way for T2I systems that can better reflect the ever-changing real world.

## 1 Introduction

Text-to-image (T2I) diffusion models have rapidly advanced, producing high-fidelity, stylistically rich images from natural-language prompts and broadening access to creative tools (Liu et al., 2024; Gao et al., 2025; Black-Forest-Labs et al., 2025). Recent models are even capable of generating more photorealistic images that adhere to common artistic conventions (Imagen-Team-Google et al., 2024; Blattmann et al., 2023). Despite this progress, a persistent failure mode remains: models frequently misinterpret prompts that reference *novel concepts*, *long-tail entities*, or *domain-specific terminology* that fall outside their pretraining distribution (Rege et al., 2025; Zhao et al., 2025). As such failure modes are manifestations of evolving world knowledge, static pretrained representations will inevitably lag behind, establishing a clear mandate for research in this direction.

Potential solutions include scaling training or fine-tuning, but it is expensive and ill-suited for rapidly emerging or long tail concepts (Li et al., 2024; Arar et al., 2024). Another solution could be optimizing the prompts rather than the model weights directly, so that the input is formulated in a way that best understood by the model. However, current prompt-optimization approaches improve image aesthetics and prompt consistency but largely operate at the text surface (Hao et al., 2022; Mañas et al., 2024). When a model lacks the underlying semantic grounding for a concept, adding descriptors like "highly detailed, 8K, award-winning" does not induce the correct depiction (Khan et al., 2025).

We propose to systematically mitigate prompt–model misalignment where the root cause is missing world knowledge, without retraining or extending the base model's capabilities directly. To this end, we employ the framework of prompt optimization and extend it as an *agentic* decision process that (i) diagnoses whether a generation failure is due to rendering limitations versus concept-comprehension failures, and (ii) conditionally invokes targeted strategies that incorporate external world knowledge. Concretely, our system (Fig. 1) integrates web interaction for evidence gathering, semantic decomposition and concept substitution for text reformulation, and multi-modal grounding via image retrieval and reference image-based conditioning. Rather than hoping the model will

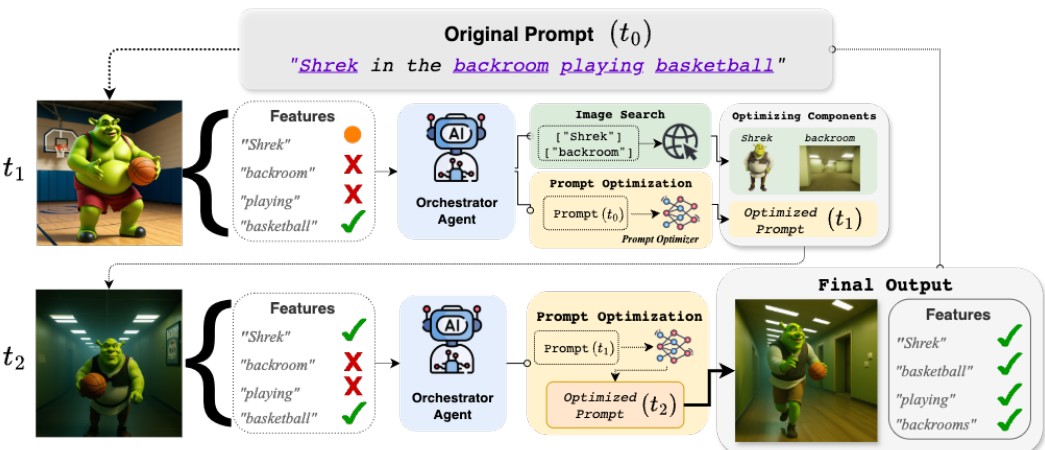

Figure 1: Overview of WORLD-TO-IMAGE.

infer unseen concepts from adjectives, the agent supplies *multimodal evidence* that steers generation toward semantically faithful outputs. By formulating the input prompt optimization as a means to instill world knowledge, we leave the base model unchanged and leverage the full potential of the existing capabilities.

Given a user prompt, the agent first conducts a lightweight failure analysis using probe generations and concept coverage checks. If signals indicate comprehension risk due to novel concepts present in the prompt, the agent retrieves concise textual definitions and representative reference images from the web, then performs: (1) *semantic decomposition* to isolate atomic concepts; (2) *concept substitution* to map obscure terms to model-familiar paraphrases while preserving meaning; and (3) *visual grounding* that conditions the generator with retrieved references.

To best study the novel/long-tail entities and compositional attributes, we curated a dataset containing prompts with novel concepts outside the training of the base model. Across popular benchmarks and our proposed dataset, our framework, W2I, consistently improves semantic faithfulness and prompt adherence over strong text-only prompt optimizers, while maintaining competitive aesthetic quality.

Our main contributions are two-fold:

1. **Agentic optimization framework.** We propose a diagnosis-and-selection agent that chooses among semantic decomposition, concept substitution, and multi-modal grounding with web-sourced evidence (Fig. 1, Sec. 3).
2. **World-knowledge infusion for T2I.** We extend prompt optimization beyond text by integrating image retrieval and conditioning to handle novel concepts, yielding state-of-the-art semantic faithfulness without retraining (Sec. 4.1).

## 2 RELATED WORKS

Prior work has explored diverse strategies, including iterative prompt optimization to emphasize salient semantic components for improved image quality, fine-tuning of model parameters to enhance generative performance, and augmentation with external knowledge sources to overcome the limitations of fixed pretrained image datasets.

### 2.1 PROMPT OPTIMIZATION IN TEXT-TO-IMAGE

Recent research has increasingly focused on automating prompt engineering to enhance the quality, control, and reliability of text-to-image (T2I) models. A dominant approach involves leveraging large language models (LLMs) and reinforcement learning to automatically discover superior prompts, optimizing for aesthetic quality and semantic alignment without requiring manual iteration. These methods range from reward-agnostic, test-time optimization in the embedding space

(Kim et al., 2025) to Multi-stage fine-tuning frameworks for LMs — multi-stage frameworks using fine-tuned Language Models (Wang et al., 2023a), and even dynamic systems that adjust prompt weights online during the generation process (Mo et al., 2024). Beyond general performance, this optimization paradigm is being extended to address critical concerns such as safety and fairness, with studies proposing universal optimizers for reliable generation (Wu et al., 2024) and techniques to improve the representation of minority groups (Um & Ye, 2025). Complementing these automated approaches, interactive systems like PromptMagician (Feng et al., 2023) focus on human-in-the-loop optimization, providing visual analytics to empower users in the creative refinement process. Collectively, this body of work signifies a shift from manual prompt crafting to systematic, goal-driven optimization frameworks for T2I synthesis.

## 2.2 World Knowledge Driven Text-to-Image

A growing body of literature has focused on creating benchmarks to probe the knowledge-grounding capabilities of T2I models. For instance, WorldGenBench (Zhang et al., 2025) introduces a benchmark to test the grounding of prompts containing explicit and implicit cultural, factual, and inferential knowledge. Using a proposed Knowledge Checklist Score, they find that while diffusion models are competent, newer autoregressive systems like GPT-4o demonstrate superior reasoning. Similarly, WISE (Niu et al., 2025) presents an extensive evaluation framework with over 1,000 prompts across 25 knowledge domains. Their WiScore metric reveals deep limitations in current models' ability to handle complex semantic, factual, and inferential concepts. Complementing these broad-knowledge benchmarks, the Commonsense-T2I challenge (Fu et al., 2024) specifically investigates whether models possess human-like commonsense reasoning. Through adversarial prompt pairs, their work highlights a significant gap between model-generated outputs and commonsense expectations, underscoring the need for improved reasoning capabilities. Collectively, these evaluation frameworks establish a clear consensus: even state-of-the-art T2I models struggle to consistently and accurately reflect nuanced world knowledge and commonsense, a gap our work aims to address.

## 3 World-To-Image: Agent-Driven World-Knowledge T2I Generation

The goal of this work is to enable T2I models to incorporate external world knowledge, thereby extending regions of the embedding space that were not observed during pretraining. Since the model has not been exposed to novel concepts during training, its performance on prompts $p$ that introduce such concepts often degrades, requiring additional time and iterations to produce meaningful images.

To address this limitation, we propose World-To-Image (W2I), an iterative, agent-based T2I generation optimization framework that dynamically utilizes world knowledge. Given an initial prompt $p_0 = p$, the system first generates a baseline image $I_0 = \text{T2I}(p_0, \phi(E_0))$ with no exemplars ($E_0 = \varnothing$). At each iteration $t$, the framework is coordinated by an Orchestrator Agent that receives the state $(p_{t-1}, I_{t-1}, E_{t-1}, s_{t-1})$, where $s_{t-1} = f(I_{t-1}, p, E_{t-1})$ is the evaluation score combining semantic alignment and aesthetic quality. Based on this state, the Orchestrator decides whether to activate the Prompt Optimizer Agent (POA) or the Image Retriever Agent (IRA).

As illustrated in Figure 2, if invoke-POA $= 1$, the POA refines the prompt $p_{t-1}$ into $p_t$ by augmenting its descriptive content (e.g., replacing domain-specific jargon or reformulating cultural references), while keeping the exemplar set unchanged ($E_t = E_{t-1}$). Conversely, if invoke-IRA $= 1$, the IRA retrieves an updated exemplar set $E_t$ conditioned on $(E_{t-1}, p_t, I_{t-1})$, grounding novel concepts such as unseen entities or styles, while leaving the prompt unchanged ($p_t = p_{t-1}$). Finally, the framework supports a joint activation where both agents operate sequentially. In this mode, the POA first generates an optimized prompt $p_t$, which is then immediately used by the IRA to retrieve a more contextually-aware set of exemplars $E_t$. This allows for a comprehensive update to both the language and vision inputs in a single iteration.

The updated prompt–exemplar pair $(p_t, E_t)$ is then passed to the generator, producing a new image $I_t = \text{T2I}(p_t, \phi(E_t))$. The image is evaluated by $s_t = f(I_t, p, E_t)$, and the loop continues until convergence. Convergence is defined either when $s_t \geq \tau$, yielding $I^* = I_t$, or when the maximum

iteration budget $T_{\max}$ is reached, in which case the best image across all iterations is returned:

$$I^* = \arg \max_{t \le T_{\max}} f(I_t, p, E_t).$$

We decompose $s_t = f(I_t, p, E_t)$ into semantic alignment, keyword coverage (graded by an LLM), and aesthetic quality:

$$s_t = \alpha\, S_t^{\text{sem}} + \beta\, K_t + \gamma\, A_t, \quad \alpha, \beta, \gamma \ge 0.$$

**Keyword set.** From the prompt $p$ (and, when applicable, reference descriptors extracted from $E_t$), we form a canonical set of required tokens $\mathcal{K} = \{k_i\}_{i=1}^m$ including entities, attributes, relations, styles, and constraints (e.g., *character name, location, palette, era, camera*). We obtain $\mathcal{K}$ by rule-based parsing (POS/NER) composed with an LLM pass that merges synonyms and prunes redundancies.

**LLM keyword grading.** An LLM receives *(prompt $p$, references $E_t$, visual analysis of $I_t$)* and returns per-keyword judgments $g_i \in \{1, \frac{1}{2}, 0\}$ for {present, partially present, missing}, with short rationales. The keyword coverage score is

$$K_t = \frac{1}{m} \sum_{i=1}^m g_i \in [0, 1],$$

optionally weighted if some keywords are marked *critical* by the Orchestrator (weights renormalized to 1).

**Aesthetic quality.** $A_t \in [0, 1]$ measures perceptual appeal (e.g., composition, lighting, color harmony). It may be computed by an automatic quality model or an LLM aesthetic rubric; scores are normalized to $[0, 1]$.

Figure 2: Illustration of a case where the Orchestrator Agent invokes the Image Retriever Agent (*invoke-IRA*=1).

In this way, we integrate both language-space optimization (via prompt refinement) and vision-space optimization (via exemplar retrieval), enabling T2I models to adapt to novel concepts during inference. We hypothesize that such a joint optimization of the language and vision space complements each other and generates a strong synergy. We formerly illustrate our method in Algorithm 1 of Appendix A.

## 4 EXPERIMENTS

This section first describes our experimental settings (4.1), then presents results analysis (4.2), aligning them with our hypotheses.

### 4.1 EXPERIMENT SETTING

**Models.** We compare seven systems: Stable Diffusion 1.4 (Rombach et al., 2022), Stable Diffusion 2.1 (Rombach et al., 2022), Stable Diffusion XL (Base) (Podell et al., 2024), OmniGen2 (Wu et al., 2025), the Promptist prompt-optimization pipeline with Stable Diffusion XL (Base) and OmniGen2 (Hao et al., 2022), and World-To-Image, our agentic pipeline.

SDXL-Base marginally outperforms OmniGen2 on general prompts (Table 1). However, in reference-conditioned settings, where prompts require grounding to unfamiliar entities or fine-grained attributes, OmniGen2 demonstrates stronger conditioning fidelity and stability, yielding higher Accuracy-to-Prompt. Accordingly, we adopt OmniGen2 as the generator backbone for our agentic pipeline, while reporting SDXL-Base, SD2.1, SD1.4, and Promptist as baselines for completeness. We include SDXL-Base, SD2.1, and SD1.4 because they remain widely adopted, strong

baselines in the image-generation community and provide a representative benchmark for comparing modern systems.

**Datasets.** To evaluate our agentic image generation pipeline, where the system issues API calls to fetch reference images for concepts the base generator is unlikely to comprehend, we use three datasets: *Lexica* (Shen et al., 2024), *DiffusionDB* (Wang et al., 2023b), and our curated *NICE* (Niche Concept Evaluation) benchmark. While existing benchmarks largely focus on generic prompts, *NICE* specifically targets rare, compositional, and time-sensitive concepts, providing a challenging setting to stress-test retrieval and grounding capabilities. For each subcategory, we searched for trending and emerging topics and refined them into high-quality prompts using GPT-5 to ensure clarity and diversity.

**General-purpose baselines.** Lexica and DiffusionDB are widely used for benchmarking text-to-image systems on broad, in-distribution prompts. While they contain occasional IP or celebrity mentions, such instances are incidental rather than the main focus of these corpora; consequently, they underrepresent the long-tail, time-sensitive, or compositional concepts our pipeline targets.

**Curated NICE Benchmark.** To stress test retrieval, we construct a 100-prompt evaluation set spanning five sub-categories: (1) Memes, (2) Real-Time News & Events, (3) Pop Culture & IP, (4) Artists/Celebrities/Influencers, and (5) Niche Concepts (20 prompts each). Prompts are built to (i) mix two distinct concepts or (ii) reference post-2024 entities and events, creating out-of-distribution cases that require external visual evidence. This design forces the Orchestrator to invoke image-retrieval via API and ground generation on retrieved exemplars.

**Evaluation Metrics.** We evaluate our retrieval-augmented, agentic pipeline on hard/niche prompts that are typically out-of-distribution for the base generator. To capture semantic faithfulness and human-perceived quality at scale, we report an LLM Grader (Hao et al., 2022) and Human Preference Rewards (Promptist Reward (Hao et al., 2022) and ImageReward (Xu et al., 2023)), and HPSv2 (Wu et al., 2023).

**LLM Grader** (Hao et al., 2022). Following (Hao et al., 2022), an LLM-based judge scores five dimensions, *Accuracy-to-Prompt*, *Creativity & Originality*, *Visual Quality & Realism*, *Consistency & Cohesion*, and *Emotional/Thematic Resonance* with an overall aggregate. This is our primary indicator of semantic alignment on rare, compositional, or time-sensitive concepts that benefit from retrieval.

**Human-Preference.** *Promptist Reward* (Hao et al., 2022) and *ImageReward* (Xu et al., 2023) are learned reward models trained on human preference data for text–image pairs; we report their sum as the Human Preference Reward. *HPSv2* (Wu et al., 2023) is another human-preference-based scoring model. These serve as automatic proxies for perceptual quality and user favorability, complementing the LLM Grader for large-scale, reproducible comparisons.

**Implementation Details.** All agents in our pipeline use `gpt-4o` as their backbone model. We perform two optimization iterations by default, using OmniGen2 as the base image generator. For image retrieval, we leverage the Google SERP API to fetch relevant reference images for grounding. The Orchestrator Agent monitors progress and may terminate the loop early if no further improvements are expected; otherwise, it executes the full two-iteration optimization schedule.

### 4.2 RESULTS

Our main results are summarized in Table 1. Across all three studied datasets, our proposed method, W2I, consistently outperforms all baselines. The overall performance gains are most significant on our curated NICE (+5.8%), compared to the broader DiffusionDB (+2.4%) and Lexica (+3.4%) benchmarks. This confirms that our agentic pipeline is particularly effective for the out-of-distribution prompts it was designed to address. The improvements are most pronounced on *Accuracy-to-Prompt*, where W2I increases the score by a substantial **+8.1%** on our set, versus +3.4% on DiffusionDB and +6.4% on Lexica. This aligns with our central hypothesis that prompts involving novel concepts benefit most from multimodal grounding, which W2I achieves by jointly leveraging retrieval and textual optimization.

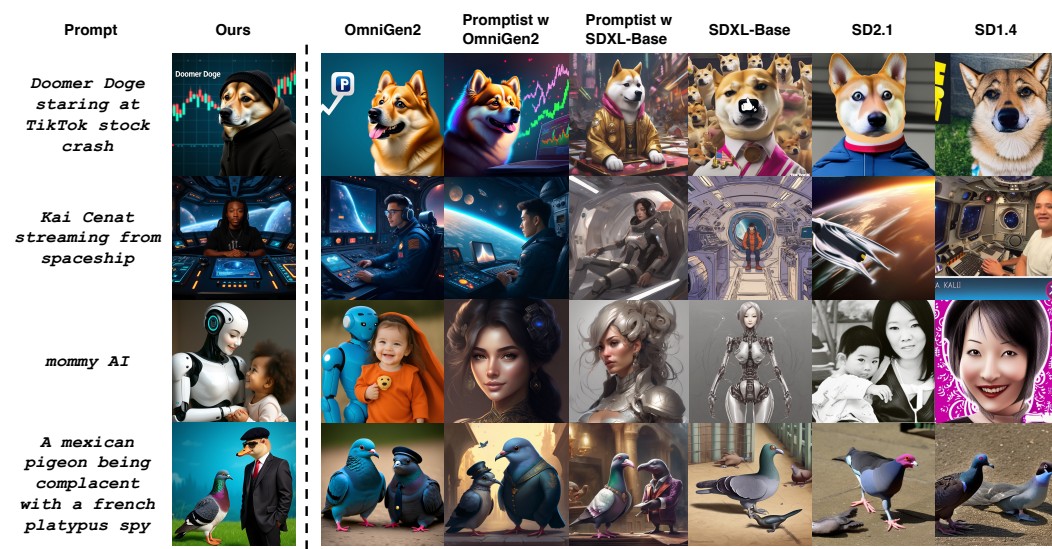

Figure 3: Qualitative comparison of text-to-image generation results across seven models. Our model consistently demonstrates stronger semantic alignment (e.g., "Doomer Doge staring at TikTok stock crash"), accurate identity grounding (e.g., "Kai Cenat streaming from spaceship"), and faithful concept representation (e.g., "mommy AI"), outperforming baselines in both fidelity and prompt adherence.

| Dataset | Metric | W2I (Ours) | OmniGen2 | Promptist w OmniGen2 | Promptist w SDXL-Base | SDXL-Base | SD2.1 | SD1.4 |
|---|---|---|---|---|---|---|---|---|
| NICE | Emotional / Thematic Resonance | **87.5** | 73.8 | 80.7 | 80.5 | 79.3 | 68.2 | 66.3 |
| | Consistency & Cohesion | **88.9** | 86.0 | 85.6 | 84.9 | 85.1 | 75.4 | 71.6 |
| | Visual Quality & Realism | **91.3** | 90.1 | 90.6 | 88.7 | 86.1 | 74.6 | 74.6 |
| | Creativity & Originality | **84.5** | 77.0 | 84.0 | 81.3 | 79.4 | 69.8 | 69.9 |
| | Accuracy-to-Prompt | **86.8** (↑ **8.1 %**) | 75.5 | 79.0 | 79.7 | 79.4 | 69.7 | 67.3 |
| | **Overall** | **87.8** (↑ **4.5 %**) | 80.5 | 84.0 | 83.0 | 81.8 | 71.5 | 69.9 |
| DiffusionDB | Emotional / Thematic Resonance | **87.3** | 81.8 | 84.6 | 84.4 | 83.7 | 77.5 | 75.8 |
| | Consistency & Cohesion | **92.3** | 89.7 | 89.8 | 88.7 | 89.4 | 82.5 | 80.0 |
| | Visual Quality & Realism | **94.1** | 92.8 | **94.1** | 93.4 | 90.0 | 81.4 | 79.2 |
| | Creativity & Originality | 85.0 | 81.6 | **86.2** | 85.0 | 83.1 | 77.1 | 76.2 |
| | Accuracy-to-Prompt | **87.4** (↑ **3.4 %**) | 81.9 | 82.8 | 84.3 | 84.5 | 79.0 | 76.9 |
| | **Overall** | **89.3** (↑ **2.1 %**) | 85.6 | 87.5 | 87.2 | 86.2 | 79.5 | 77.6 |
| Lexica | Emotional / Thematic Resonance | **88.6** | 81.6 | 86.1 | 85.2 | 83.4 | 76.9 | 75.9 |
| | Consistency & Cohesion | **92.7** | 90.3 | 90.3 | 89.2 | 88.0 | 82.5 | 81.7 |
| | Visual Quality & Realism | **95.2** | 94.2 | 93.3 | 93.1 | 89.2 | 83.0 | 79.1 |
| | Creativity & Originality | **86.3** | 79.8 | 85.5 | 85.2 | 83.4 | 77.7 | 76.4 |
| | Accuracy-to-Prompt | **89.8** (↑ **6.0 %**) | 83.6 | 84.7 | 84.4 | 83.8 | 79.4 | 77.0 |
| | **Overall** | **90.5** (↑ **2.8 %**) | 85.9 | 88.0 | 87.5 | 85.6 | 79.9 | 78.0 |

Table 1: Comparison of LLM-based evaluation metrics across datasets and models. Bold values indicate the best performance within each dataset group. For our main metrics, Accuracy-to-Prompt and Overall, we additionally report the relative improvement (in %) over the next best-performing model within the same dataset group.

| Dataset | Metric | W2I (Ours) | OmniGen2 | Promptist w OmniGen2 | Promptist w SDXL-Base | SDXL-Base | SD2.1 | SD1.4 |
|---|---|---|---|---|---|---|---|---|
| NICE | Human Preference Reward | **2.761** | 2.259 | 2.4040 | 2.156 | 2.005 | 1.609 | 1.305 |
| | ImageReward | **1.271** | 0.775 | 0.8119 | 0.601 | 0.550 | 0.239 | -0.022 |
| | HPSv2 | **0.296** | 0.283 | 0.2815 | 0.278 | 0.278 | 0.256 | 0.243 |
| DiffusionDB | Human Preference Reward | **2.817** | 2.364 | 2.6854 | 2.331 | 2.233 | 1.639 | 1.409 |
| | Image Reward | **1.271** | 0.993 | 1.0357 | 0.695 | 0.696 | 0.224 | 0.033 |
| | HPSv2 | **0.304** | 0.297 | 0.2977 | 0.286 | 0.281 | 0.252 | 0.241 |
| Lexica | Human Preference Reward | **2.947** | 2.738 | 2.8673 | 2.420 | 2.303 | 1.647 | 1.528 |
| | Image Reward | **1.376** | 1.176 | 1.2208 | 0.766 | 0.766 | 0.210 | 0.122 |
| | HPSv2 | **0.309** | 0.302 | 0.2998 | 0.287 | 0.283 | 0.247 | 0.241 |

Table 2: Comparison of Human-Preference evaluation metrics across datasets and models. Bold values indicate the best performance within each dataset group.

**Image Quality and Human Preference**  In Table 2, we study the impact of our multi-modal prompt optimization on image quality. We focus on both objective image quality scores and human preference-based evaluations. As shown, W2I maintains strong performance across both dimensions, outperforming all other baselines. These findings indicate that our method does not sacrifice visual fidelity in pursuit of semantic accuracy, but instead achieves a strong balance between the two.

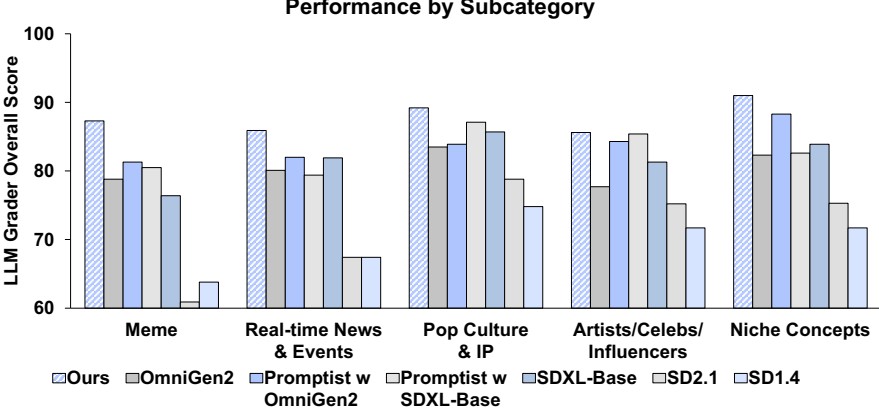

Figure 4: LLM Grader overall scores across subcategories. Our method consistently outperforms all baselines.

**Performance on Novel Concepts**  To further validate our framework's effectiveness with out-of-distribution prompts, we analyzed its performance across the five distinct subcategories of our NICE benchmark. As illustrated in Figure 4, our method consistently outperforms all baselines, including the strong *Promptist* optimizer and the base *OmniGen2* model, in each category—from memes and real-time events to niche intellectual property. This result demonstrates the framework's robustness and confirms that its superior performance is driven by a specialized ability to handle a wide range of previously unseen concepts through agentic retrieval and grounding.

| Metric | W2I (Ours) | Prompt Optimizer | Image Retrieval | w/o Agent |
|---|---|---|---|---|
| Emotional / Thematic Resonance | **87.5** | 74.6 | 79.3 | 73.8 |
| Consistency & Cohesion | **88.9** | 85.3 | 84.2 | 86.0 |
| Visual Quality & Realism | **91.3** | 89.8 | 89.1 | 90.1 |
| Creativity & Originality | **84.5** | 76.6 | 80.6 | 77.0 |
| Accuracy-to-Prompt | **86.8** | 76.4 | 79.7 | 75.5 |
| Overall Score | **87.8** | 80.5 | 82.6 | 80.5 |
| Human Preference Reward | **2.761** | 2.624 | 2.319 | 2.259 |
| ImageReward | **1.271** | 1.098 | 0.853 | 0.775 |
| HPSv2 | 0.296 | **0.299** | 0.288 | 0.283 |

Table 3: Ablation study on our dataset. Each column shows performance when a specific component is removed to quantify its contribution. Prompt Optimizer indicates that only the Prompt Optimizer (with image retrieval disabled) was used. Image Retrieval indicates that only the Image Retrieval module was used. w/o Agent represents a variant with no agents. Bold values indicate the best performance.

**Ablation Study**  To disentangle the contributions of different components within our optimization pipeline, we coablated each component of the optimization pipeline (Table 3). Across the board, our full pipeline yields the best results on our proposed dataset. Relying exclusively on image retrieval can fail for more complex prompts, as the generation process may become overly conditioned on the reference without fully aligning to the task specification. Conversely, *prompt optimization only* improves alignment with textual instructions but image conditioning can provide the model with a more concrete reference. The synergy of combining both components produces significant gains

across all metrics, indicating that while each method individually emphasizes different axes of improvement, only their combination unlocks the full potential of the base model.

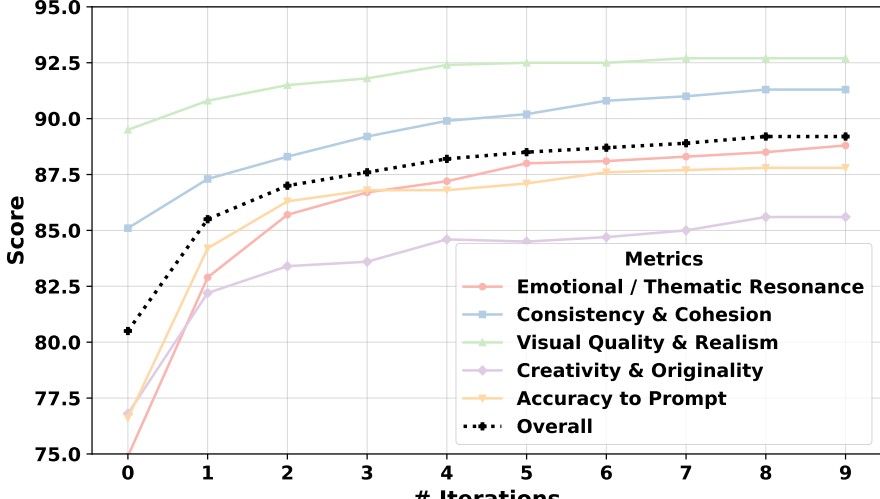

Figure 5: LLM-Grader sub-scores and overall score across optimization steps. The dotted line shows the overall score; solid lines represent individual dimensions.

**Impact of increasing optimization steps**  We also analyzed the impact of extending the optimization schedule up to 10 steps, and plot the per-iteration improvement traces in Figure 5. Performance improves consistently across iterations, with the sharpest increase shown in the first 2 iterations. This supports our decision to use 2-step iterations by default, striking a balance between performance and efficiency. We also observe that IRA is often invoked in the early iterations and POA predominantly in the later iterations, suggesting that image retrieval provides a strong early boost, while subsequent prompt optimization refines outputs for further gains.

## 5  DISCUSSIONS

Our findings raise several important discussion points. The strong gains on *novel concepts* highlight that pretrained generative models often already possess latent capacity to represent new entities, but require the right multimodal signals to activate them. This suggests a broader opportunity: instead of scaling models alone, improving interface mechanisms, such as retrieval and adaptive prompting, may unlock substantial gains. Moreover, our ablation study shows a strong synergy between text and image-based optimization, effectively expanding the horizon of prompt optimization to multimodal prompts to harness their complementary strengths.

Future work may explore the scalability of World-To-Image with respect to the number of novel concepts in the input prompt. Preliminary results suggest that World-To-Image can consolidate novel concepts from various sources (text, image) and effectively incorporate the knowledge into the generation process through multiple iterations, which may lead to its ability of compositional generalization over multiple novel concepts simultaneously.

While W2I demonstrates consistent improvements, several limitations remain. First, the reliance on external image retrieval assumes access to relevant, high-quality references; in domains with sparse or noisy imagery, performance may degrade. Second, our method focuses on optimizing prompts and retrieval rather than modifying the base generative model, which means it cannot introduce fundamentally new capabilities. This is a compromise that in turn enabled a efficient and model-agnostic framework, which lets us leverage the capabilities of the base model otherwise locked due to the limitations in the prompt comprehension. Finally, iterative optimization introduces additional test-time computational overhead compared to single-pass baselines, while our framework provides a flexible control knobs to balance the efficiency-quality trade-off.

## 6 CONCLUSION

In this work, we present World-To-Image, an agentic framework that solves the prompt-model mismatch problem rooted in missing world knowledge by multimodal prompt optimization. By deploying an Orchestrator agent to dynamically select between language-space prompt refinement and vision-space visual grounding via web retrieval, W2I instills timely world knowledge into the generation process without modifying the base model. Our experiments demonstrated that this approach significantly outperforms existing methods, achieving a +8.1% improvement in accuracy-to-prompt on our challenging NICE benchmark containing diverse novel concepts.

Our findings provide evidence that the path toward more capable generative models lies not only in model size scaling but also in improving the interfaces. The strong performance of W2I shows that by dynamically searching external knowledge and instilling them through multimodal interface, we can unlock the latent capabilities of existing models and bridge the gap between their static training and the evolving world. As a result, World-To-Image introduces a new axis of improvement for T2I generation, while also providing a flexible framework for future research into more efficient retrieval strategies and more sophisticated agentic reasoning.

## ETHICS STATEMENT

We recognize that powerful text-to-image models, including our framework, can be misused to generate misinformation, harmful stereotypes, and explicit content. Our web-retrieval mechanism introduces two main concerns: propagation of societal biases from search engine algorithms and potential copyright issues when conditioning on web-sourced images, particularly for protected characters or artist styles. Our work focuses on the agentic optimization mechanism and preserves all safety filters of the backbone model (OmniGen2), and we recommend that future implementations use ethically-sourced, licensed, or public-domain retrieval corpora. This research aims to advance multimodal AI reasoning for positive, creative applications.

## REPRODUCIBILITY STATEMENT

To ensure the reproducibility of our results, we provide the following details regarding our experimental setup. All resources are available in the supplementary material and will be released upon publication.

**Code:** The code for our agentic framework, including the implementation of the Orchestrator, Prompt Optimizer, and Image Retrieval agents, will be made publicly available at `https://github.com/anonym-code996/World-To-Image`.

**Models:** Our agents use `gpt-4o` as the backbone model. The core generative model is `OmniGen2`. Baselines include `Stable Diffusion 1.4, 2.1, SDXL-Base`, and the `Promptist` pipeline applied to both `OmniGen2` and `SDXL-Base`. All models were used with their publicly available weights.

**Datasets:** We evaluate our framework on two public benchmarks, **Lexica** and **DiffusionDB**, as well as our curated **NICE** benchmark. The prompts for the NICE benchmark will be included in the code repository.

**APIs and Services:** The Image Retrieval Agent utilizes the Google Search Engine Results Page (SERP) API for sourcing reference images.

**Evaluation:** All evaluation was conducted using publicly available models and codebases. LLM-Grader scores were obtained following the methodology of Hao et al. (2022). Human Preference scores were calculated using the official `ImageReward` and `HPSv2` models. The specific prompts and generated images used for evaluation are included in the supplementary material.

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

# A  WORLD-TO-IMAGE ALGORITHM

---

**Algorithm 1:** World-To-Image: Agentic Framework for Optimizing Novel-Concept T2I Generation

---

**Legend.**

- $p$: initial user prompt.
- $p_t$: refined prompt at iteration $t$.
- $I_t$: generated image at iteration $t$.
- $I^*$: final selected image.
- $E_t$: set of external exemplars (retrieved reference images) at iteration $t$.
- $\phi(E_t)$: embedding/conditioning function applied to exemplars.
- $f(I_t, p, E_t)$: evaluation function (e.g., LLMGrader, CLIP similarity, or aesthetic score).
- $\tau$: score threshold for convergence.
- $T_{\max}$: maximum iteration budget.
- OrchestratorAgent: decides whether to invoke sub-agents.
- PromptOptimizerAgent: refines/augments prompts.
- ImageRetrieverAgent: retrieves external exemplars.
- invoke-POA, invoke-IRA: binary flags from the Orchestrator.

**Input:** Initial prompt $p$; threshold $\tau$; maximum iterations $T_{\max}$
**Output:** Final image $I^*$
$p_0 \leftarrow p,\ E_0 \leftarrow \varnothing$ ;
$I_0 \leftarrow \text{T2I}(p_0, \phi(E_0))$ ;
**for** $t \leftarrow 1$ **to** $T_{\max}$ **do**
    // Step 1:  Orchestration
    $(\text{invoke-POA}, \text{invoke-IRA}) \leftarrow \text{OrchestratorAgent}(p_{t-1}, I_{t-1}, E_{t-1})$ ;
    // Step 2:  Prompt Optimization (if selected)
    **if** *invoke-POA* $= 1$ **then**
        $p_t \leftarrow \text{PromptOptimizerAgent}(p_{t-1}, I_{t-1})$
    **else**
        $p_t \leftarrow p_{t-1}$
    // Step 3:  Image Retrieval (if selected)
    **if** *invoke-IRA* $= 1$ **then**
        $E_t \leftarrow \text{ImageRetrieverAgent}(E_{t-1}, p_t, I_{t-1})$
    **else**
        $E_t \leftarrow E_{t-1}$
    // Step 4:  Candidate Generation & Scoring
    $I_t \leftarrow \text{T2I}(p_t, \phi(E_t))$ ;
    $s_t \leftarrow f(I_t, p, E_t)$ ;
    **if** $s_t \geq \tau$ **then**
        $I^* \leftarrow I_t$ ;
        **break**
**return** $I^*$

---

## B EXTENDED VISUAL COMPARISONS

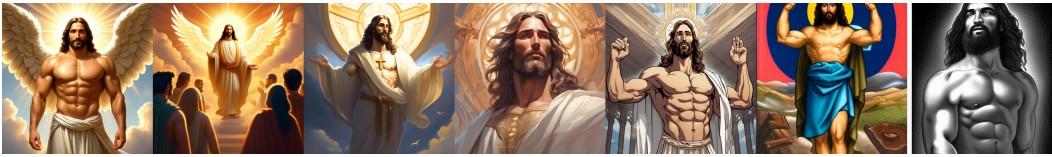

buff jesus christ welcoming you into heaven

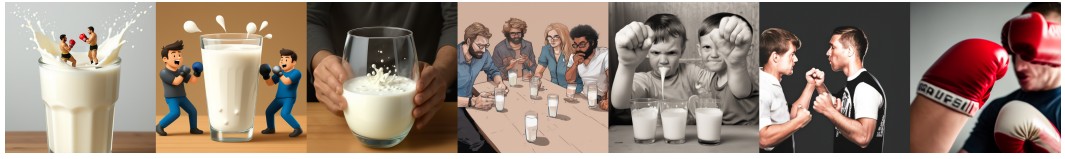

punching contest in a glass of milk

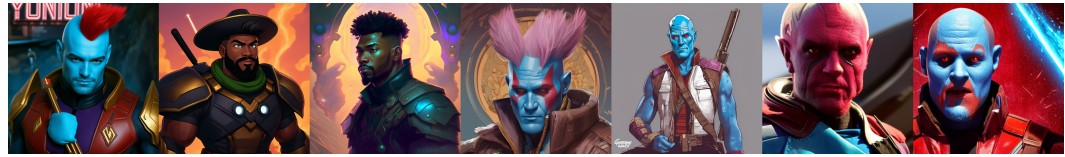

Yondu udonta from guardians of the galaxy

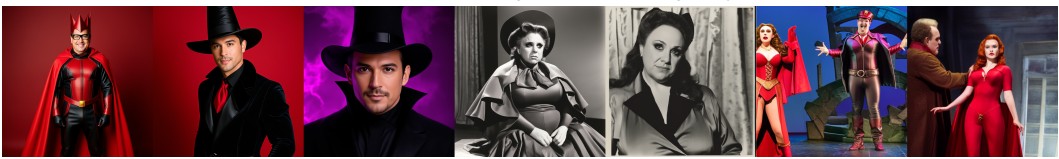

production photo of Danny Divito as the Scarlett witch

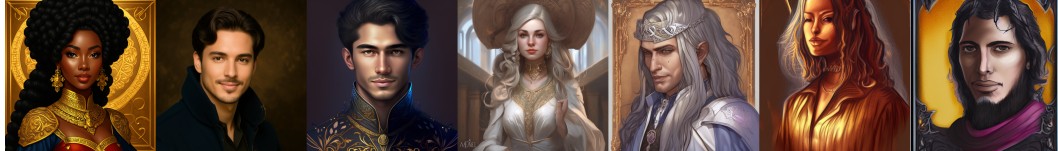

portrait of mel medarda from arcane.

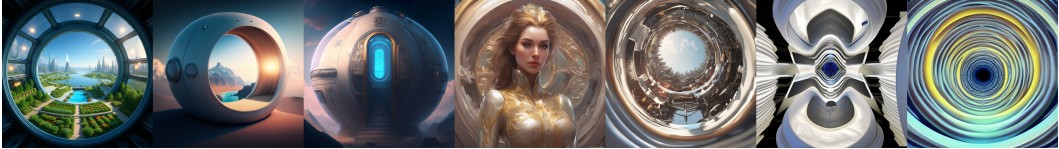

a beautiful concept inside of a o'neill cylinder

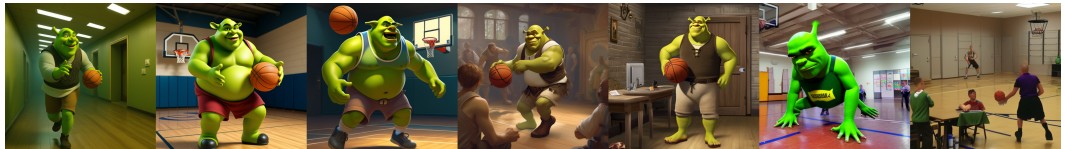

Shrek in the Backrooms playing basketball

Figure 6: Qualitative comparison of image generations across models for diverse prompts. Each row corresponds to one prompt, with columns showing outputs from left to right: Ours, OmniGen2, Promptist w OmniGen2, Promptist w SDXL-Base, SDXL-Base, SD2.1, and SD1.4.

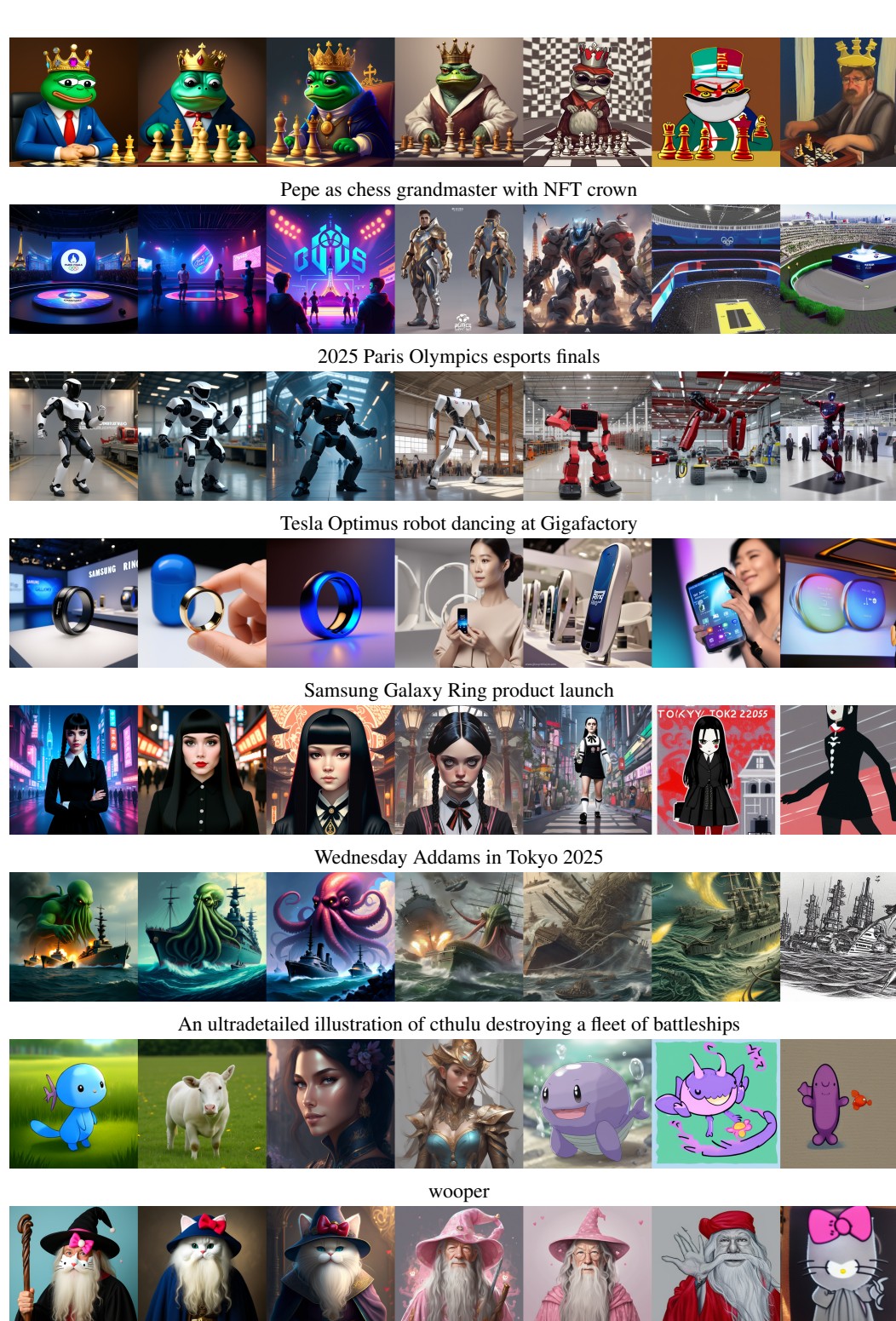

Figure 7: Qualitative comparison of image generations across models for diverse prompts. Each row corresponds to one prompt, with columns showing outputs from left to right: Ours, OmniGen2, Promptist w OmniGen2, Promptist w SDXL-Base, SDXL-Base, SD2.1, and SD1.4.

## C  EXTENDED TABLE

| Metric | W2I(Ours) | OmniGen2 | Promptist w OmniGen2 | Promptist w SDXL-Base | SDXL-Base | SD2.1 | SD1.4 |
|---|---|---|---|---|---|---|---|
| Emotional / Thematic Resonance | **86.0** | 68.0 | 73.0 | 73.5 | 70.0 | 51.0 | 52.6 |
| Consistency & Cohesion | **91.0** | 87.5 | 85.0 | 84.0 | 82.5 | 70.0 | 70.0 |
| Visual Quality & Realism | 89.0 | 91.0 | **92.0** | 90.5 | 83.0 | 71.5 | 72.1 |
| Creativity & Originality | **83.5** | 75.0 | **83.5** | 81.5 | 77.0 | 58.5 | 64.7 |
| Accuracy-to-Prompt | **87.0** | 72.5 | 73.0 | 73.0 | 69.5 | 54.0 | 59.5 |
| Overall | **87.3** | 78.8 | 81.3 | 80.5 | 76.4 | 60.9 | 63.8 |
| Human Preference Reward | **3.032** | 2.750 | 2.783 | 2.377 | 2.033 | 1.309 | 0.958 |
| ImageReward | **1.546** | 1.279 | 1.172 | 0.732 | 0.582 | -0.008 | -0.315 |
| HPSv2 | **0.313** | 0.305 | 0.299 | 0.280 | 0.268 | 0.252 | 0.236 |

Table 4: Comparison of Scores for Meme subgroup across different models. The best scores are highlighted in bold.

| Metric | W2I(Ours) | OmniGen2 | Promptist w OmniGen2 | Promptist w SDXL-Base | SDXL-Base | SD2.1 | SD1.4 |
|---|---|---|---|---|---|---|---|
| Emotional / Thematic Resonance | **85.0** | 75.0 | 77.9 | 78.5 | 80.5 | 65.0 | 65.0 |
| Consistency & Cohesion | 86.5 | 83.5 | **86.8** | 81.5 | 86.5 | 69.5 | 68.5 |
| Visual Quality & Realism | **92.5** | 88.5 | 89.5 | 85.0 | 84.5 | 69.5 | 71.5 |
| Creativity & Originality | **79.0** | 75.5 | 78.4 | 75.5 | 76.5 | 65.5 | 66.0 |
| Accuracy-to-Prompt | **86.5** | 78.0 | 77.4 | 78.0 | 81.5 | 67.5 | 66.0 |
| Overall | **85.9** | 80.1 | 82.0 | 79.4 | 81.9 | 67.4 | 67.4 |
| Human Preference Reward | **2.615** | 1.712 | 2.131 | 1.632 | 1.846 | 1.628 | 1.264 |
| ImageReward | **1.179** | 0.297 | 0.636 | 0.210 | 0.429 | 0.309 | -0.047 |
| HPSv2 | **0.284** | 0.258 | 0.266 | 0.252 | 0.265 | 0.245 | 0.229 |

Table 5: Comparison of Scores for Real-time News & Events subgroup across different models. The best scores are highlighted in bold.

| Metric | W2I(Ours) | OmniGen2 | Promptist w OmniGen2 | Promptist w SDXL-Base | SDXL-Base | SD2.1 | SD1.4 |
|---|---|---|---|---|---|---|---|
| Emotional / Thematic Resonance | **90.5** | 80.0 | 83.0 | 88.0 | 87.5 | 81.5 | 76.0 |
| Consistency & Cohesion | **89.5** | 87.5 | 83.5 | 87.5 | 84.0 | 77.5 | 73.5 |
| Visual Quality & Realism | **95.0** | 91.5 | 89.5 | 90.0 | 89.0 | 78.0 | 75.5 |
| Creativity & Originality | 81.5 | 77.5 | 83.0 | **83.5** | 82.0 | 75.5 | 73.5 |
| Accuracy-to-Prompt | **89.5** | 81.0 | 80.5 | 86.5 | 86.0 | 81.5 | 75.5 |
| Overall | **89.2** | 83.5 | 83.9 | 87.1 | 85.7 | 78.8 | 74.8 |
| Human Preference Reward | **2.218** | 1.974 | 1.981 | 2.017 | 1.735 | 1.610 | 1.310 |
| ImageReward | **0.712** | 0.441 | 0.375 | 0.471 | 0.276 | 0.219 | -0.017 |
| HPSv2 | **0.280** | 0.270 | 0.264 | 0.276 | 0.273 | 0.259 | 0.240 |

Table 6: Comparison of Scores for the Pop Culture & IP subgroup across different models. The best scores are highlighted in bold.

| Metric | W2I(Ours) | OmniGen2 | Promptist w OmniGen2 | Promptist w SDXL-Base | SDXL-Base | SD2.1 | SD1.4 |
|---|---|---|---|---|---|---|---|
| Emotional / Thematic Resonance | **85.5** | 69.5 | 82.0 | 84.2 | 77.9 | 73.0 | 70.5 |
| Consistency & Cohesion | **88.0** | 84.5 | 85.5 | 86.3 | 85.8 | 78.5 | 71.0 |
| Visual Quality & Realism | 89.0 | 89.0 | **91.0** | 89.4 | 85.8 | 75.5 | 75.0 |
| Creativity & Originality | **84.0** | 76.0 | 83.0 | 83.2 | 76.3 | 75.5 | 73.5 |
| Accuracy-to-Prompt | 81.5 | 69.5 | 80.0 | **83.7** | 80.5 | 73.5 | 68.5 |
| Overall | **85.6** | 77.7 | 84.3 | 85.4 | 81.3 | 75.2 | 71.7 |
| Human Preference Reward | **2.826** | 2.187 | 2.511 | 2.276 | 2.050 | 1.740 | 1.549 |
| ImageReward | **1.344** | 0.707 | 0.885 | 0.711 | 0.617 | 0.354 | 0.239 |
| HPSv2 | **0.293** | 0.277 | 0.282 | 0.295 | 0.289 | 0.263 | 0.257 |

Table 7: Comparison of Scores for the Artists, Celebrities, Influencers subgroup across different models. The best scores are highlighted in bold.

| Metric | W2I(Ours) | OmniGen2 | Promptist w OmniGen2 | Promptist w SDXL-Base | SDXL-Base | SD2.1 | SD1.4 |
|---|---|---|---|---|---|---|---|
| Emotional / Thematic Resonance | **90.5** | 76.5 | 87.5 | 78.5 | 80.5 | 70.5 | 66.5 |
| Consistency & Cohesion | **89.5** | 87.0 | 87.0 | 85.5 | 86.5 | 81.5 | 75.0 |
| Visual Quality & Realism | **91.0** | 90.5 | 91.0 | 88.5 | 88.0 | 78.5 | 79.0 |
| Creativity & Originality | **94.5** | 81.0 | 92.0 | 83.0 | 85.0 | 74.0 | 71.5 |
| Accuracy-to-Prompt | **89.5** | 76.5 | 84.0 | 77.5 | 79.5 | 72.0 | 66.5 |
| **Overall** | **91.0** | 82.3 | 88.3 | 82.6 | 83.9 | 75.3 | 71.7 |
| Human Preference Reward | **3.115** | 2.670 | 2.60 | 2.480 | 2.362 | 1.757 | 1.443 |
| ImageReward | **1.574** | 1.150 | 0.982 | 0.883 | 0.844 | 0.322 | 0.029 |
| HPSv2 | **0.312** | 0.308 | 0.296 | 0.290 | 0.292 | 0.262 | 0.251 |

Table 8: Comparison of Scores for Niche Concepts subgroup across different models. The best scores are highlighted in bold.

| Metric | NICE | | DiffusionDB | | Lexica | |
|---|---|---|---|---|---|---|
| | W2I | OmniGen2 | W2I | OmniGen2 | W2I | OmniGen2 |
| Promptist Reward | **-0.143** | -0.285 | **-0.178** | -0.259 | **-0.117** | -0.210 |
| Aesthetic Score | **5.961** | 5.936 | **6.184** | **6.184** | **6.284** | 6.246 |

Table 9: Comparison of Promptist Reward and Aesthetic Score across our model and OmniGen2 on three datasets. Lower is better for Promptist Reward; higher is better for Aesthetic Score.

| Metric | Meme | Real-Time News & Events | Pop Culture & IP | Artists, Celebrities, Influencers | Niche Concept |
|---|---|---|---|---|---|
| Emotional / Thematic Resonance | 86.0 | 85.0 | 90.5 | 85.5 | 90.5 |
| Consistency & Cohesion | 91.0 | 86.5 | 89.5 | 88.0 | 89.5 |
| Visual Quality & Realism | 89.0 | 92.5 | 95.0 | 89.0 | 91.0 |
| Creativity & Originality | 83.5 | 79.0 | 81.5 | 84.0 | 94.5 |
| Accuracy-to-Prompt | 87.0 | 86.5 | 89.5 | 81.5 | 89.5 |
| Overall Score | 87.3 | 85.9 | 89.2 | 85.6 | 91.0 |
| Human Preference Reward | 3.032 | 2.615 | 2.218 | 2.826 | 3.115 |
| ImageReward | 1.546 | 1.179 | 0.712 | 1.344 | 1.574 |
| HPSv2 | 0.313 | 0.284 | 0.279 | 0.293 | 0.312 |

Table 10: Performance of World-To-Image on NICE benchmark subgroups. We report LLM-Grader and Human Preference metrics.

# D    PROMPT TEMPLATES

## D.1    ORCHESTRATOR AGENT

---
**Orchestrator Agent**

You are an expert orchestrator for multimodal generation model.

Your job is to:

1. Analyze the provided image, prompt, scores, and optimization history.

2. Decide the most suitable generation task type: (This is in order of preference)

- **text_image_to_image**: Use a reference image + prompt for improved fidelity. (Most recommended)

- **text_to_image**: Generate image purely from text prompt.

- **image_editing_with_prompt_and_reference**: Modify the currently generated image according to the prompt and reference image.

- **image_editing_with_prompt**: Modify the currently generated image according to the prompt (inpainting, style transfer, attribute edit).

### GUIDELINES

- Image editing is the least recommended task type. - You should only choose image editing if the generated image is very good and you are confident that the prompt is not enough to improve the image.

### INPUTS

Original Prompt: {original_prompt}

Current Opimtized Prompt: {current_prompt}

Detailed Scores: {json.dumps(current_scores, indent=2)}

Optimization History: {json.dumps(optimization_history, indent=2)}

Visual Analysis: {visual_analysis}

### TASK CLASSIFICATION RULES

- **text_to_image**: Prompt is self-sufficient; no celebrity/IP likeness, no niche style, no need to preserve an existing image.

- **text_image_to_image**: Prompt includes niche entities (celebrity/IP/meme), rare styles, or ambiguous visuals → retrieve TWO references.

- **image_editing_with_prompt**: A previously generated image exists AND the new text indicates incremental change (style tweak, color, local edit) without needing a specific external reference.

- **image_editing_with_prompt_and_reference**: A previously generated image exists AND the new text implies matching a specific look/scene/face/style from a known IP or example → retrieve ONE reference.

### DISAMBIGUATION (TEXT-ONLY PROMPTS THAT MIGHT BE EDITS)

- If OPTIMIZATION_HISTORY shows a recent successful generation (e.g., within last step) and DETAILED_SCORES indicate high content alignment but style mismatch → prefer **image_editing_with_prompt**. - If the text asks to match a specific world/IP/location/face (e.g., 'Shrek swamp', 'Monica's apartment', 'Van Gogh brushwork') → prefer **image_editing_with_prompt_and_reference**. - If structural changes are large (pose/layout/object count), or prior image is low-quality/incorrect content → prefer **text_image_to_image** (with references if niche) or **text_to_image**. - Reference needed should just be a simple keyword or a list of keywords.

### STRATEGY SELECTION

- **text_to_image** → ['prompt_optimizer']

- **text_image_to_image** → ['prompt_optimizer', 'image_retrieval']

- **image_editing_with_prompt** → ['prompt_optimizer']

- **image_editing_with_prompt_and_reference** → ['prompt_optimizer', 'image_retrieval']
---

### Output Format

Return a JSON object:
```
{
  'task_type': 'text_to_image' | 'text_image_to_image' |
'image_editing_with_prompt' | 'image_editing_with_prompt_and_reference',
  'strategies': ['prompt_optimizer', 'image_retrieval'],
  'references_needed': ['reference_image_1', 'reference_image_2'],
  'draft_prompt': 'Draft prompt for the prompt optimizer to optimize
with reference image index not _REF.',
  'reasoning': 'Step-by-step reasoning why this task type and
strategies were chosen.',
  'score_analysis': 'Interpretation of each score and threshold
violations.',
  'keyword_analysis': 'Which keywords are crucial/missing and how
they influence strategy choice.',
  'confidence': 0.0
}
```

### Few Shot Examples

FEW-SHOT EXAMPLES

EXAMPLE 1 (TEXT_IMAGE_TO_IMAGE; HARD IP)

Prompt: 'Squid Game S3 teaser poster, Gi-hun in a rain-soaked street, neon green mask reflections'
Output:
```
{
  'task_type': 'text_image_to_image',
  'strategies': ['prompt_optimizer', 'image_retrieval'],
  'references_needed': ['squid game poster', 'gi-hun'],
  'draft_prompt': 'The poster based on image 1, a man from image 2
in a rain-soaked street, neon green mask reflections',
  'reasoning': 'IP + character likeness + specific aesthetic → needs
two references (Gi-hun, official poster style) to anchor identity
and tone.',
  'score_analysis': 'clip_score low; face_similarity target absent;
style_consistency uncertain → retrieval to ground likeness/style.',
  'keyword_analysis': ''Squid Game', 'Gi-hun', 'neon mask' are
niche; require grounding.',
  'confidence': 0.93
}
```

EXAMPLE 2 (TEXT_TO_IMAGE; GENERIC BUT DESCRIPTIVE)

Prompt: 'Pixel art of a golden retriever surfing a giant wave at sunset'
Output:
```
{
  'task_type': 'text_to_image',
  'strategies': ['prompt_optimizer'],
  'references_needed': [],
  'draft_prompt': 'Pixel art of a golden retriever surfing a giant
wave at sunset',
  'reasoning': 'No niche entities; text fully specifies subject,
action, style.',
  'score_analysis': 'semantic_alignment expected adequate; no prior
image constraints.',
  'keyword_analysis': ''pixel art', 'retriever', 'surfing', 'sunset'
are common.',
  'confidence': 0.90
}
```

---

**Few Shot Examples (cont.)**

EXAMPLE 3 (IMAGE_EDITING_WITH_PROMPT; TEXT-ONLY PROMPT BUT EDIT PRIOR IMAGE)

Context: A valid image was just generated (step t-1) of 'street portrait, female runner mid-stride'.
Prompt (text-only): 'Give it a 90s VHS sitcom vibe with warm halation and grain; keep the same pose and outfit'
Output:

```
{
  'task_type':  'image_editing_with_prompt',
  'strategies':  ['prompt_optimizer'],
  'references_needed':  [],
  'draft_prompt':  'Give it a 90s VHS sitcom vibe with warm halation
and grain; keep the same pose and outfit',
  'reasoning':  'Text suggests incremental style change to the most
recent image while preserving pose/outfit.  No specific external
reference required.',
  'score_analysis':  'prior_image_available=true;
content_alignment_high=0.86; style_mismatch=0.41;
edit_intent_detected=true → style-only edit is appropriate.',
  'keyword_analysis':  ''90s VHS', 'grain', 'halation' are style
modifiers without named IP → no retrieval.',
  'confidence':  0.95
}
```

EXAMPLE 4 (IMAGE_EDITING_WITH_PROMPT_AND_REFERENCE; TEXT-ONLY PROMPT BUT NEEDS IP/BACKGROUND MATCH)

\# The original image will always be image 1. And there will be only one reference image which is image 2.
\# Only retrieve one reference image.
Context: A valid image was just generated (step t-1) of 'ogre-like character standing in a forest clearing'.
Prompt (text-only): 'Keep the current pose and lighting but move her to the Shrek swamp and match the movie's green tint and fog'
Output:

```
{
  'task_type':  'image_editing_with_prompt_and_reference',
  'strategies':  ['prompt_optimizer', 'image_retrieval'],
  'references_needed':  ['shrek'],
  'draft_prompt':  'Keep the current pose and lighting but move her
to the green ogre in image 1 and match the movie's green tint and
fog',
  'reasoning':  'User wants to retain existing composition but match
a specific IP location and look.  External visual target needed for
accurate palette/props/fog.',
  'score_analysis':  'prior_image_available=true;
content_alignment_high=0.83; location_specificity='Shrek swamp';
style_target='movie's green tint' → requires one reference to lock
scene aesthetics.',
  'keyword_analysis':  ''Shrek swamp', 'movie's green tint', 'fog' →
IP-scene keywords necessitate reference.',
  'confidence':  0.96
}
```

## D.2   PROMPT OPTIMIZER AGENT

---

**Prompt Optimizer Agent**

ROLE

You are the Prompt Optimizer Agent. Rewrite the user's request into a clean, actionable instruction string for the selected task type. Always produce a single JSON object with the following variables:

- A single string variable named `prompt`
- A `negative_prompts` comma-separated string

TASK TYPE

{`task_type`}

INPUTS

- ORIGINAL PROMPT: {`original_prompt`}
- CURRENT OPTIMIZED PROMPT: {`current_prompt`}
- VISUAL ANALYSIS: {`visual_analysis`}
- CURRENT SCORES: {`score_summary`}
- RECENT OPTIMIZATION HISTORY: {`history_block`}
- ORCHESTRATOR REASONING: {`reasoning`}

OBJECTIVES

- Preserve essential subject(s), action/intent, and any crucial style/medium cues.
- If there are any unclear or ambiguous concepts that the image generator might not know try explaining them in the prompt.
- Clarify composition, lighting, lens/camera, time-of-day only when helpful.
- Keep wording compact, natural, and non-contradictory.
- Append concise negatives if artifacts are likely (e.g., 'no text artifacts, natural hands').
- If a concept is niche/ambiguous (celebrity, brand, rare object/place/style)
- Always refer to the reference image(s) with image index in the prompt for higher performance.

---

---

**Prompt Optimizer Agent**

OUTPUT RULES (CHOOSE EXACTLY ONE CASE BASED ON TASK_TYPE)

**A) text_to_image**

```
{
  'prompt':  '<refined prompt string>',
  'negative_prompts':  'term1, term2, term3',
}
```

*Guidelines:*

- One complete directive (Subject → Action/Intent → Composition → Lighting/Camera → Style/Medium).
- Rich but controlled descriptors; avoid long enumerations or conflicting specs.

**B) text_image_to_image**

```
{
  'prompt':  '<composite instruction referencing the reference(s)>',
  'negative_prompts':  'term1, term2, term3'
}
```

*Guidelines:*

- Assume the Image Retrieval Agent provides reference image(s) for the niche concept(s).
- Instruction should state the intended composition/edit/compositing with those references.
- For example 'Add the cat in image 1 to the background in image 2.'
- Always refer to the reference image(s) with image index in the prompt for higher performance.

**C) image_editing_with_prompt**

```
{
  'prompt':  '<instruction to improve the current image>',
  'negative_prompts':  'term1, term2, term3',
}
```

**D) image_editing_with_prompt_and_reference**

```
{
  'prompt':  '<instruction to improve using reference(s)>',
  'negative_prompts':  'term1, term2, term3',
}
```

*Guidelines for Image Editing:*

- You're improving an EXISTING image to better match the SAME prompt
- Analyze what's wrong with current image (from scores/visual analysis)
- For prompt-only editing: focus on lighting, color, style, composition improvements
- For reference editing: identify specific elements that need external reference
- Keep the core subject/scene but improve quality/accuracy

STYLE HEURISTICS

- Prioritize: Subject → Action/Intent → Composition → Lighting/Camera → Style/Medium.
- Use concrete, photography/film/art vocabulary over vague adjectives.
- Avoid contradictions (e.g., 'harsh noon sun' + 'soft night ambience').
- If scores/history imply distortions, add short negatives (hands, faces, watermarks, banding, text).

```
Few Shot Examples
```

CASE A: TEXT_TO_IMAGE

Original propmt: 'Sunrise garden macro photography'

```
{
  'prompt':  'The sun rises slightly; clear dew on rose petals; a
crystal ladybug crawls toward a dew bead; early-morning garden
backdrop; macro lens.',
  'negative_prompts':  '(((deformed))), blurry, over saturation, bad
anatomy, disfigured, poorly drawn face, mutation, mutated,
(extra_limb), (ugly), (poorly drawn hands), fused fingers, messy
drawing, broken legs censor, censored, censor_bar'
}
```

CASE B1: TEXT_IMAGE_TO_IMAGE

Original prompt: 'Dr Strange in backroom'

```
{
  'prompt':  'Compose a scene with the character (Dr Strange) from
image 1 standing in a dim, fluorescent 'backrooms' corridor from
image 2; center-frame, medium shot; flat overhead lighting, subtle
fog; emphasize iconic outfit and cape motion.',
  'negative prompts':  'text artifacts, over-smoothing, waxy skin,
warped hands, banding'
}
```

CASE B2: TEXT_IMAGE_TO_IMAGE

Original prompt: 'A kid's toy in a parking lot.' {

```
  'prompt':  'Place the toy from image 1 into the hands of the person
in image 2 in a parking-lot setting; align scale and grip; match
lighting direction and color temperature.',
  'negative prompts':  '(((deformed))), blurry, over saturation, bad
anatomy, disfigured, poorly drawn face, mutation, mutated,
(extra_limb), (ugly), (poorly drawn hands), fused fingers, messy
drawing, broken legs censor, censored, censor_bar'
}
```

CASE C: IMAGE_EDITING_WITH_PROMPT

Original prompt: 'Dr Strange in backroom'
Current image issues: Low lighting quality, poor color balance

```
{
  'prompt':  'Improve the lighting and color balance of the current
character (Dr Strange) in backroom scene; enhance contrast and fix
dim areas; maintain character pose and backroom atmosphere',
  'negative prompts':  'overexposure, harsh shadows, color banding,
washed out colors',
}
```

CASE D: IMAGE_EDITING_WITH_PROMPT_AND_REFERENCE

Original prompt: 'Dr Strange in backroom'
Current image issues: Character face doesn't look like Dr Strange

```
{
  'prompt':  'Fix the character's face in the current backroom scene
to match image 2 (character (Dr Strange)); maintain the existing
pose and backroom setting in image 1; improve facial accuracy',
  'negative_prompts':  'wrong face, generic face, blurry features,
face artifacts',
}
```

**Note:** Emit exactly one case per call based on task type. No extra text outside the JSON object.

## D.3 IMAGE RETRIEVAL AGENT

**Image Retrieval Agent**

You are an expert visual analyst evaluating reference images for text-to-image generation.

CONTEXT:

- Original prompt: {original_prompt}
- Search query: {query}
- Category: {category}
- Purpose: Select the best reference images to guide AI image generation.
- You must select at least one image.

TASK:
Analyze each provided image and evaluate how well it matches the search query and would help generate the target prompt.
For {category} category:
- CONTENT: Look for objects, people, locations, compositions that match the query
- STYLE: Look for artistic styles, visual aesthetics, color palettes, techniques
- CONTEXT: Look for environmental context, mood, atmosphere, setting details

EVALUATION CRITERIA:
1. **Query Match**: How well does the image match the specific search query?
2. **Visual Quality**: Is the image clear, well-composed, and visually appealing?
3. **Usefulness**: Would this image provide good visual guidance for AI generation?
4. **Distinctiveness**: Does it offer unique visual information not found in other candidates?

INSTRUCTIONS:
- Rate each image from 0.0 to 1.0 (higher = better)
- Select up to {max_selections} best images
- Provide brief reasoning for each selection

Respond with ONLY a JSON object in the following format (this is an example):
```
{
  'selections': [
    {
      'image_index': 0,
      'score': 0.85,
      'reasoning': 'Excellent match for query, high visual quality,
provides clear guidance'
    },
    {
      'image_index': 1,
      'score': 0.72,
      'reasoning': 'Good secondary option with different
angle/perspective'
    }
  ]
}
```

Only include images you would actually select (score $\geq 0.6$).
If you are not sure about the images, you can select multiple images. Low scores are allowed.

---

**Query Rewriting Prompt**

You are an expert at creating image search queries. A search query failed to return any images from an image search API.

CONTEXT:

- Original text prompt: 'original prompt'
- Failed search query: 'original query'
- Goal: Find reference images to help generate the target prompt

TASK:
Create a better, more searchable query that is likely to return relevant images.
Consider:

- **Simplify complex terms**: Replace uncommon/specific terms with more common alternatives
- **Add descriptive keywords**: Include visual descriptors that would help find relevant images
- **Use popular terms**: Replace niche concepts with mainstream equivalents
- **Consider synonyms**: Use alternative words that mean the same thing
- **Focus on visual elements**: Emphasize what the image should look like rather than abstract concepts

EXAMPLES:

- "Dr Strange" → "Marvel superhero with cape" or "sorcerer with magic"
- "backroom" → "yellow fluorescent office space" or "liminal empty rooms"
- "cyberpunk hacker" → "futuristic computer user neon lights"
- "medieval knight" → "armored warrior with sword"

Respond with ONLY the modified search query, nothing else. Make it 2-6 words that would likely return relevant images.

---

**Visual Analysis Prompt**

You are an expert at analyzing images and detecting AI-generated artifacts. Provide concise, focused analysis.
Analyze this image and compare it with the text: 'prompt'.
Focus on:
1) What the text describes well vs. what it misses
2) Any hallucinations or distorted details that don't match the prompt.
3) Any elements that are not shown in the text but should be added.
4) Visual enhancements for better generation quality
Be specific about enhancement opportunities that don't conflict with the original intent.

## D.4 IMAGE RETRIEVAL AGENT

---

**LLM Grader Prompt**

You are a multimodal large-language model tasked with evaluating images generated by a text-to-image model. Your goal is to assess each generated image based on specific aspects and provide a detailed critique, along with a scoring system. The final output should be formatted as a JSON object containing individual scores for each aspect and an overall score.

**1. Key Evaluation Aspects and Scoring Criteria:**

For each aspect, provide a score from 0 to 10 (0 = poor, 10 = excellent) and a short justification (1–2 sentences).

- **Accuracy to Prompt** – Assess how well the image matches the prompt: elements, objects, and setting.
- **Creativity and Originality** – Judge whether the image shows imagination beyond a literal interpretation.
- **Visual Quality and Realism** – Evaluate resolution, detail, lighting, shading, and perspective.
- **Consistency and Cohesion** – Check whether all elements are coherent and free from anomalies.
- **Emotional or Thematic Resonance** – Assess whether the image conveys the intended mood or tone.

**2. Overall Score:** Provide an overall score as a weighted or simple average of all aspects.

Please evaluate the following image based on the prompt: `"{prompt}"`

Respond with a JSON object in this exact format:

```
{
    "accuracy_to_prompt": {
        "score": <0-10>,
        "explanation": "<1-2 sentence explanation>"
    },
    "creativity_and_originality": {
        "score": <0-10>,
        "explanation": "<1-2 sentence explanation>"
    },
    "visual_quality_and_realism": {
        "score": <0-10>,
        "explanation": "<1-2 sentence explanation>"
    },
    "consistency_and_cohesion": {
        "score": <0-10>,
        "explanation": "<1-2 sentence explanation>"
    },
    "emotional_or_thematic_resonance": {
        "score": <0-10>,
        "explanation": "<1-2 sentence explanation>"
    },
    "overall_score": <0-10>
}
```

---

