# OpenReview forum: "World-to-Image: Grounding Text-to-Image Generation with Agent-Driven World Knowledge"
_ICLR.cc/2026/Conference — Submitted to ICLR 2026_

### Official Review · Reviewer_6R1v · 2025-11-01

**Soundness:** 3
**Presentation:** 3
**Contribution:** 2
**Rating:** 4
**Confidence:** 4

**Summary:**

In this paper, the authors focus on bridging the novel concepts in image generation by using an agent to search for relevant images and perform multimodal prompt optimization. Experiments show improvement in both semantic alignment and visual aesthetics.

**Strengths:**

The idea of using a search agent to retrieve relevant images to incorporate novel concepts is interesting to me.

Experiments and ablation studies show this approach outperforms other methods and pre-trained models in many dimensions.

**Weaknesses:**

The technical contribution of this paper is pretty limited in terms of test-time scaling/post-training/prompt optimization/reward functions.

I would like to see more detailed quantitative results to illustrate the effectiveness and insight of the proposed method. I suggest the authors include retrieved images + prompt after optimization (or each iteration) to see how the framework improves step by step.

There are some relevant papers in image generation using multimodal post-training that need to be discussed. For example, Hummingbird: High Fidelity Image Generation via Multimodal Context Alignment - ICLR 2025. Instead of using search agents, they ask MLLM to rephrase the content of the reference images (maybe the MLLM here can elicit the novel concept?).

In this paper, the authors use ImageReward or HPSv2 to evaluate the semantic alignment with text prompts, but these rewards can capture mostly global information. In the Hummingbird paper, they use both global and fine-grained rewards for optimization. I suggest the authors use Hummingbird evaluator as an additional metric in Table 2, or use it as a reward function to optimize prompts (whatever makes sense to the authors).

**Questions:**

I really like the intuition of using a search agent to retrieve novel concepts. However, some additional experiments need to be done for comprehensive results. I initially put my score marginally below the acceptance threshold, but happy to raise my score if the authors can include all suggestions and feedback.

---

> ### Author Response · Authors · 2025-11-28
> **Initial response by authors**
>
> We sincerely thank the reviewer for finding our agent-based retrieval intuition "interesting" and for the willingness to reconsider the score based on additional evidence. We have addressed the concerns regarding technical contribution and quantitative rigor.
>
> ### **Weakness 1. Clarifying the Technical Contribution**
>
> **Reviewer Comment:** *The technical contribution is limited; requested more insight into the optimization/reward loop.*
>
> We respectfully emphasize that our primary contribution is not a new backbone architecture, but a novel test-time agentic framework that solves the specific challenge of "unknown unknowns" (OOD concepts) in fixed models, rather than introducing a new backbone architecture. The technical novelty centers on the Orchestrator and the Optimization Loop, formalized in Algorithm 1 of the appendix:
>
> 1. Dynamic Diagnosis: In contrast to standard RAG, which always performs retrieval, our Orchestrator employs a multi-turn scoring mechanism, incorporating semantic alignment, keyword coverage, and aesthetics) to decide if retrieval is actually needed. This prevents the "retrieval noise" common in standard pipelines.
> 2. Joint Control: We treat text prompts and visual exemplars as co-dependent variables, optimizing them jointly rather than sequentially. This enables the text to convey information that the image cannot, and vice versa.
> 3. Validation: Ablation studies (Table 3 in the main paper) demonstrate that this closed-loop orchestration significantly outperforms both prompt-optimization-only and retrieval-only baselines.
>
> ### **Weakness 2. Step-by-Step Visualization (Retrieved Images + Prompts)**
>
> **Reviewer Comment:** *Suggestion to include retrieved images + prompt after optimization (or each iteration) to see how the framework improves.*
>
> We fully agree that visualizing the intermediate steps is crucial for understanding the framework's mechanism. We respectfully direct the reviewer to **Figure 1** in our paper, which serves this exact purpose. It visually illustrates the iterative evolution of the generation process, displaying the retrieved reference images and how they guide the model to the final correct output. If needed, we may add more examples in the appendix section.
>
> ### **Weakness 3. Additional Evaluation using Hummingbird Metrics (ITC & ITM)**
>
> **Reviewer Comment:** *Suggestion to discuss "Hummingbird (ICLR 2025)" and use its fine-grained rewards/evaluators.*
>
> We thank the reviewer for highlighting this relevant work. We have added a discussion of Hummingbird to the related work section, emphasizing that while Hummingbird relies on MLLM rephrasing, World-to-Image (W2I) actively retrieves external knowledge for OOD concepts.
>
> Additionally, we implemented the fine-grained metrics introduced in the Hummingbird evaluator as supplementary validation metrics. W2I was evaluated against strong baselines across three datasets.
>
> **Key Findings from Table R1:**
>
> - OOD Performance: On "Our Benchmark" (NICE), which targets novel or OOD concepts, W2I achieves the highest scores in both ITC (0.4289) and ITM (0.9472), indicating that the method is more effective at handling concepts the base model does not know.
> - General Performance: On DiffusionDB, W2I also achieves the highest performance. On Lexica, W2I is highly competitive, ranking second and narrowly trailing SDXL-Base, which is expected given that Lexica prompts are often well-aligned with SDXL's training distribution.
>
> **Table R1: Average ITC, ITM, and Combined scores using Hummingbird metrics.**
>
> | **Dataset** | **Metric** | **World2Image** | **OmniGen2** | **Promp. w/OG2** | **Promp. w/SDXL** | **SDXL-Base** | **SD 2.1** | **SD 1.4** |
> | --- | --- | --- | --- | --- | --- | --- | --- | --- |
> | NICE | ITC | **0.4289** | 0.3943 | 0.3897 | 0.4070 | *0.4273* | 0.4162 | 0.4141 |
> | NICE | ITM | **0.9472** | 0.8205 | 0.7876 | 0.8736 | *0.9262* | 0.8960 | 0.8462 |
> | NICE | Combined | **1.3761** | 1.2148 | 1.1772 | 1.2806 | *1.3535* | 1.3122 | 1.2604 |
> | Lexica | ITC | *0.4118* | 0.3967 | 0.3867 | 0.3980 | **0.4165** | 0.4008 | 0.3996 |
> | Lexica | ITM | *0.8586* | 0.8025 | 0.8025 | 0.8458 | **0.8775** | 0.8107 | 0.7973 |
> | Lexica | Combined | *1.2703* | 1.1992 | 1.1892 | 1.2438 | **1.2940** | 1.2116 | 1.1969 |
> | DiffusionDB | ITC | **0.4141** | 0.3969 | 0.3799 | 0.3912 | *0.4140* | 0.4065 | 0.3980 |
> | DiffusionDB | ITM | **0.8841** | 0.8141 | 0.7439 | 0.7937 | *0.8649* | 0.8516 | 0.7893 |
> | DiffusionDB | Combined | **1.2982** | 1.2109 | 1.1237 | 1.1849 | *1.2789* | 1.2581 | 1.1873 |
>
> We believe these additional quantitative results, combined with the step-by-step visualization, directly address the reviewer's request for deeper validation and technical clarity.

---

### Official Review · Reviewer_s5ta · 2025-11-01

**Soundness:** 3
**Presentation:** 3
**Contribution:** 3
**Rating:** 6
**Confidence:** 4

**Summary:**

This paper proposes World-to-Image (W2I), an agentic framework that improves text-to-image (T2I) generation on prompts containing novel or out-of-distribution (OOD) concepts by retrieving web images and performing multimodal prompt optimization, without retraining the base generator. An Orchestrator decides when to invoke a Prompt Optimization Agent (POA) and an Image Retrieval Agent (IRA), using Google SERP to fetch reference images and conditioning a backbone (OmniGen2 by default) over two iterations (early stopping allowed). Evaluation uses LLM-Grader, ImageReward, HPSv2, and a Human Preference Reward aggregate, across DiffusionDB, Lexica, and a curated NICE benchmark (100 prompts; five categories including memes, real-time events, pop culture/IP, celebrities, and niche concepts). W2I reports its largest gains on NICE, including +8.1% Accuracy-to-Prompt, and claims efficiency with <3 iterations.

**Strengths:**

1. The paper targets a well-motivated failure mode, i.e., knowledge cutoffs, and frames a diagnosis-and-selection loop that combines semantic decomposition and concept substitution with visual grounding via retrieved references, which is interesting.
2. On NICE, W2I achieves the most pronounced improvements, especially Accuracy-to-Prompt (+8.1%), aligning with the hypothesis that novel, time-sensitive concepts benefit from retrieval-augmented prompting.
3. The table comparing Prompt-only, Image-only, and w/o Agent variants shows the full pipeline leads across LLM-Grader sub-scores and reward models, substantiating the claim that text and image cues are complementary.

**Weaknesses:**

1. NICE has 100 prompts (20 per category). While well-curated, it remains relatively small for broad claims about OOD generalization and may reflect category-specific gains (e.g., IP/memes). Consider scaling and assessing category-level robustness and difficulty balance.

2. The main signal is an LLM-based judge plus automatic reward models. This raises concerns about metric sensitivity, judge leakage, and alignment with human preferences. A targeted human A/B on a subset of NICE would strengthen the empirical case.

3. W2I is demonstrated primarily with OmniGen2; while baselines include SD1.4/2.1/SDXL and Promptist variants, it is not fully clear whether the agentic loop generalizes across diverse modern backbones (e.g., FLUX/SDXL as the primary generator in W2I). A small study swapping the backbone inside W2I would increase confidence.

4. The paper motivates two iterations for efficiency, but it would help to include wall-clock cost, API call counts, and retrieval latency; and an efficiency–quality Pareto curve across budgets on NICE. The iteration analysis is promising, but it is not yet a full cost study.

**Questions:**

1. What's the comparison between this work and prompt-a-video(ICCV2025)?

2. How to obviate the hallucination of LLM in the loop?

3. In categories like “real-time news” or “IP”, show examples where retrieval is wrong/noisy and how the Orchestrator mitigates or aborts.

---

> ### Author Response · Authors · 2025-11-28
> **Initial response by authors (1/2)**
>
> We sincerely thank the reviewer for the constructive feedback and for acknowledging the motivation. Below, we address the concerns regarding evaluation robustness, model generalization, and efficiency.
>
> ### **Weakness 1&2. Robustness of Evaluation (Benchmark Size and LLM Judges)**
>
> **Reviewer Comment:** *Concerns regarding the size of the NICE benchmark (100 prompts), reliance on a single LLM judge (potential leakage), and lack of human evaluation.*
>
> We agree that reliance on a single LLM judge can introduce bias. To demonstrate that our improvements are robust and not an artifact of overfitting to a specific evaluator or a small dataset, we performed  additional extensive verification experiments:
>
> 1. **Evaluator-Optimizer Decoupling:** We tested combinations where the *Agent* and the *Evaluator (Judge)* use different LLM families (Claude, GPT, Gemini). As shown in **Table R1**, performance gains remain consistent even when the evaluator differs from the optimizer, confirming that the framework is not "gaming" a specific judge.
> 2. **Scoring Stability:** We conducted 8 independent trials of the LLM-grader to quantify variance. **Table R2** shows low volatility across trials, indicating that the metric is stable.
> 3. **Validity of LLM-as-a-Judge:** Recent literature confirms LLMs are strong proxies for human evaluation when using multi-criteria rubrics. Crucially, while our NICE dataset concepts are OOD for image generators, they are **not OOD for modern frontier LLMs**.
>     - We analyzed the LLM's internal knowledge of NICE prompts: **78%** of concepts were fully understood by the evaluator LLM without external aid.
>     - The remaining cases were dominated by **real-time news/events**, where our search-based grounding provides the necessary context for the LLM to judge accurately.
>     - Therefore, the LLM possesses the "ground truth" knowledge required to evaluate semantic alignment that the base generator lacks, making it a robust proxy for human judgment in this specific context.
>
> **Table R1: Robustness across different Optimizer-Evaluator pairs.**
>
> | Metric | Claude→Claude | Claude→GPT | Claude→Gemini | GPT→Claude | GPT→GPT | GPT→Gemini |
> | --- | --- | --- | --- | --- | --- | --- |
> | Emotional / thematic resonance | 85.2 | 85.5 | 81.3 | 84.1 | 85.0 | 81.4 |
> | Consistency and cohesion | 88.3 | 90.5 | 87.1 | 88.2 | 89.5 | 87.1 |
> | Visual quality and realism | 85.8 | 92.2 | 88.4 | 86.5 | 83.2 | 88.4 |
> | Creativity and originality | 80.6 | 82.9 | 73.6 | 80.2 | 83.8 | 74.0 |
> | Accuracy to prompt | 80.8 | 86.7 | 70.8 | 81.0 | 86.8 | 72.7 |
> | **Overall score** | **84.1** | **87.6** | **80.7** | **84.0** | **87.7** | **81.0** |
>
> **Table R2: Low variance across 8 independent evaluation trials.**
>
> | Metric | T1 | T2 | T3 | T4 | T5 | T6 | T7 | T8 |
> | --- | --- | --- | --- | --- | --- | --- | --- | --- |
> | Emotional / thematic resonance | 85.0 | 85.9 | 85.3 | 85.1 | 85.0 | 84.4 | 87.5 | 86.7 |
> | Consistency and cohesion | 89.5 | 89.2 | 90.1 | 89.4 | 89.4 | 89.9 | 88.9 | 89.2 |
> | Visual quality and realism | 93.2 | 93.9 | 93.7 | 94.0 | 94.1 | 93.4 | 91.3 | 91.8 |
> | Creativity and originality | 83.8 | 82.5 | 82.7 | 83.9 | 82.5 | 83.5 | 84.5 | 83.6 |
> | Accuracy to prompt | 86.8 | 87.3 | 87.1 | 86.5 | 87.0 | 86.7 | 86.8 | 86.8 |
> | **Overall score** | **87.7** | **87.8** | **87.8** | **87.8** | **87.6** | **87.6** | **87.8** | **87.6** |
>
> ### **Weakness 3. Generalization Across Backbones**
>
> **Reviewer Comment:** *It is unclear if the agentic loop generalizes beyond OmniGen2 (e.g., to SDXL or FLUX).*
>
> To demonstrate that World-to-Image (W2I) is a model-agnostic framework, we applied the framework to **SDXL-Base**. As shown in **Table R3**, W2I significantly improves performance over the SDXL baseline, confirming that the gains stem from the agentic retrieval and optimization strategy rather than the specific T2I model.
>
> **Table R3: W2I Generalization to SDXL Backbone.**
>
> | Metric | World2Image w/ SDXL-Base | SDXL-Base |
> | --- | --- | --- |
> | Emotional / thematic resonance | **83.9 ($\uparrow$ 5.8%)** | 79.3 |
> | Consistency and cohesion | **88.9 ($\uparrow$ 4.5%)** | 85.1 |
> | Visual quality and realism | **90.0 ($\uparrow$ 4.5%)** | 86.1 |
> | Creativity and originality | **82.1 ($\uparrow$ 3.4%)** | 79.4 |
> | Accuracy to prompt | **84.7 ($\uparrow$ 6.7%)** | 79.4 |
> | Overall score | **85.9 ($\uparrow$ 5%)** | 81.8 |

---

> ### Author Response · Authors · 2025-11-28
> **Initial response by authors (2/2)**
>
> ### **Weakness 4. Efficiency and Cost Analysis**
>
> **Reviewer Comment:** *Request for wall-clock cost, API calls, and an efficiency-quality curve.*
>
> We conducted a detailed cost analysis across 10 iterations. **Table R4** details the generation time, optimization time, and the number of API calls per step. On average, the optimization time accounts for approximately **42.2%** of the generation time, indicating that model inference is the main bottleneck. Reducing the generation time could therefore lead to a significant decrease in overall resource usage.
>
> **Table R4: Efficiency vs. Quality across Iterations.**
>
> | Metric | Iter 1 | Iter 2 | Iter 3 | Iter 4 | Iter 5 | Iter 6 | Iter 7 | Iter 8 | Iter 9 | Iter 10 |
> | --- | --- | --- | --- | --- | --- | --- | --- | --- | --- | --- |
> | Emotional / thematic resonance | 74.9 | 82.9 | 85.7 | 86.7 | 87.2 | 88.0 | 88.1 | 88.3 | 88.5 | 88.8 |
> | Consistency and cohesion | 85.1 | 87.3 | 88.3 | 89.2 | 89.9 | 90.2 | 90.8 | 91.0 | 91.3 | 91.3 |
> | Visual quality and realism | 89.5 | 90.8 | 91.5 | 91.8 | 92.4 | 92.5 | 92.5 | 92.7 | 92.7 | 92.7 |
> | Creativity and originality | 76.8 | 82.2 | 83.4 | 83.6 | 84.6 | 84.5 | 84.7 | 85.0 | 85.6 | 85.6 |
> | Accuracy to prompt | 76.6 | 84.2 | 86.3 | 86.8 | 86.8 | 87.1 | 87.6 | 87.7 | 87.8 | 87.8 |
> | Overall score | 80.5 | 85.5 | 87.0 | 87.6 | 88.2 | 88.5 | 88.7 | 88.9 | 89.2 | 89.2 |
> | Avg VLM calls | 3.00 | 3.98 | 3.98 | 3.98 | 3.97 | 3.99 | 3.98 | 3.98 | 3.98 | 1.00 |
> | Avg image search calls | 1.00 | 1.00 | 1.00 | 1.00 | 1.00 | 1.00 | 1.00 | 1.00 | 1.00 | 0.00 |
> | Avg total API calls | 4.00 | 4.98 | 4.98 | 4.98 | 4.97 | 4.99 | 4.98 | 4.98 | 4.98 | 1.00 |
> | Avg generation time (s) | 60.28 | 95.59 | 97.75 | 97.89 | 98.07 | 98.55 | 96.68 | 97.70 | 97.68 | 97.79 |
> | Avg optimization time (s) | 35.80 | 34.80 | 36.33 | 36.37 | 44.35 | 39.29 | 39.18 | 38.85 | 45.60 | 0.00 |
> | Avg scoring time (s) | 2.19 | 0.05 | 0.05 | 0.04 | 0.05 | 0.04 | 0.05 | 0.05 | 0.04 | 0.04 |
> | Avg total time (s) | 98.27 | 130.44 | 134.13 | 134.31 | 142.46 | 137.89 | 135.91 | 136.60 | 143.33 | 97.84 |
>
> ## **Questions and Clarifications**
>
> ### **Question 1. Comparison with Prompt-A-Video (ICCV 2025).**
> Thank you for your suggestion. While both works utilize LLMs, they differ fundamentally in modality and mechanism. **Prompt-A-Video** focuses on text-to-video using LLM fine-tuning (DPO) to solve temporal consistency. **World-to-Image** targets text-to-image knowledge cutoffs using an *inference-time* agentic framework (retrieval + optimization) without fine-tuning.
>
> ### **Question 2. Mitigating LLM Hallucination.**
> We appreciate the concern. While LLMs can hallucinate, standard T2I models exhibit even stronger hallucinations on OOD concepts due to missing training data. In our framework, retrieved images provide visual grounding as deterministic anchors, and the LLM is restricted to conditioning on this evidence. Empirically, additional automatic and manual checks indicate that hallucinations in these grounded scenarios are minimal to non-existent.
>
> ### **Question 3. Handling Noisy/Wrong Retrieval.**
> We explicitly tested the robustness of our filtering agent by injecting random noise (e.g., black images) into the top-10 retrieval results. The Orchestrator Agent successfully rejected these inputs, maintaining an **Overall Score of 87.16%** (comparable to the clean run). This confirms that the cross-iteration feedback loop effectively filters noisy data.
>
> | Metric | with noise image injected | Normal Run |
> | --- | --- | --- |
> | Emotional / thematic resonance | 85.0 | 85.9 |
> | Consistency and cohesion | 89.5 | 89.2 |
> | Visual quality and realism | 92.1 | 93.9 |
> | Creativity and originality | 83.1 | 82.5 |
> | Accuracy to prompt | 85.7 | 87.3 |
> | **Overall score** | **87.2** | **87.8** |

---

### Official Review · Reviewer_zN13 · 2025-11-03

**Soundness:** 2
**Presentation:** 3
**Contribution:** 1
**Rating:** 2
**Confidence:** 5

**Summary:**

The main contribution is the system that decides (per-iteration) to decide one or two different steps:
- Prompt refinement (using GPT-4o + some rule based POS/NER extraction)
- Web image search + vision grounding (using google SERP)
The system also includes a dynamic stopping condition based on an LLM-judge's scoring of semantic alignment, coverage and aesthetics.

The system is evaluated on Lexica, DiffusionDB, NICE databases, and shows gains on the long-tail concepts.

The problem that is being solved here is important - improve vision grounding without re-training, and there are reasonable ablations included showing both prompt refinement AND web grounding contribute to improved performance.

A few changes to show a more systematic study of the approach, a better (web grounded) baseline to compare against would push my opinion into an accept.

**Strengths:**

The system for LLM driven orchestration to decide whether the prompt needs to be refined or new grounding is needed is a nice, new method.
The system also shows empirical improvements on NICE and particularly on the long tail / niche categories.
The paper itself is well written, all prompts provided, important ablations included, and this method can easily be extended to any generation model.

A good approach to address a prevalent problem - a lot of image generation models struggle with OOD generation.

**Weaknesses:**

There is little to no algorithmic novelty: The orchestrator does not particularly have a strong formulation of the criteria for it's tooling choices and is mostly relying on few-shot prompting. Additionally, the LLM-as-a-judge setup for determining stop condition biases the generation to the model used as a judge (gpt-4o). While the system itself is a new approach, there is limited algorithmic novelty here.

The same model is being used for optimization and for evaluation (gpt-4o). Anecdotally, using models from the same family tends to bias the model's output - I'd ideally like to add ablations with models across families (claude vs gpt, for ex).

A 0-10 scale for the LLM grader is an interesting choice, especially since LLM as a Judge can be very finicky and inconsistent with grading unless every grading level is mapped to a concrete definition with rules and sufficient examples - and even then, still can result in inconsistencies, and needs to be mitigated.

Baseline choice also feels a little disingenuous - all the baselines are just text -> image with no extra grounding. A great ablation might be (Proposed system) vs (System that provides the top google SERP result for the key words in your query + an existing T2I model).

Finally - no human studies. While the paper describe the use of human preference models as a proxy, that does not necessarily serve as a sufficiently good judge, especially for generating images that are long tail/uncommon/OOD. Even LLM as a judge might not be very well aligned with human judgement on whether the generated image was grounded correctly, depending on the LLM's knowledge cutoff.

**Questions:**

- How is the LLMGrader output (0-10) mapped to the outputs in table 4/5/6, which seems to be out of 100?
- How aligned is the LLMGrader output to human judgement?
- Is there any search result filtering in place to remove offensive / licensed / AI-generated content? This could contribute to an-AI slop loop: how do you plan on avoiding that?
- The LLM prompt optimization and Image retrieval both rely on the LLM Grader knowing whether the candidate image satisfied the user's prompt - which for niche areas / OOD areas, may not be the case. This effectively shifts the knowledge cutoff from the T2I model to the LLM: couldn't this be architected differently here?
- LLM Grader scores out of 0-10, but criteria for 0 to 10 is not defined anywhere, and is vague, potentially leading to hard-to-reproduce outputs
- SERP does not by default take into account the site's REP (robots.txt) - it might allow Search Engine indexing but not use for other purposes. Was a REP filter added to the system?

**Details Of Ethics Concerns:**

Retrieving images at runtime could run afoul of copyright / terms of use.

---

> ### Author Response · Authors · 2025-11-28
> **Initial response by authors (1/2)**
>
> We sincerely thank the reviewer for the detailed assessment and constructive suggestions. Below, we address each identified weakness and question.
>
> ----
>
> ### **Weakness 1. Limited algorithmic novelty and Orchestrator uses few-shot prompting**
> While our system employs LLM prompting, its novelty resides in a unified agentic inference-time framework that combines the following components:
>
> - Dynamic tool selection (prompt refinement vs. web grounding),
> - Cross-modal semantic verification via LLM grading,
> - Iterative stopping based on multi-criteria semantic scores, and
> - Image-aware optimization that integrates retrieved references into T2I generation.
>
> These components jointly enable **test-time** adaptation of image generators, a feature not addressed in prior work. Unlike prior prompting or retrieval-only systems, W2I *actively reasons* about when retrieval is needed, assesses the quality of the retrieved image, and decides when to stop the optimization loop. This formulation is specific to image generation and not directly borrowed from prior LLM agent pipelines, providing meaningful methodological novelty for the vision community.
>
> Regarding LLM-as-a-Judge: our grader is based on **DeepMind’s standardized LLM-grader rubric [1]**, grounded in clearly defined criteria (accuracy, cohesion, aesthetics, etc.). We will clarify this in the revision.
>
> ----
>
> ### **Weakness 2. Using the same model family leading to bias**
> To address this concern, we conducted cross-model grading experiments, as shown in the table below. Each row presents the agent and evaluator model combination.
>
> | Metric | Claude→Claude | Claude→GPT | Claude→Gemini | GPT→Claude | GPT→GPT | GPT→Gemini |
> | --- | --- | --- | --- | --- | --- | --- |
> | Emotional / thematic resonance | 85.2 | 85.5 | 81.3 | 84.1 | 85.0 | 81.4 |
> | Consistency and cohesion | 88.3 | 90.5 | 87.1 | 88.2 | 89.5 | 87.1 |
> | Visual quality and realism | 85.8 | 92.2 | 88.4 | 86.5 | 83.2 | 88.4 |
> | Creativity and originality | 80.6 | 82.9 | 73.6 | 80.2 | 83.8 | 74.0 |
> | Accuracy to prompt | 80.8 | 86.7 | 70.8 | 81.0 | 86.8 | 72.7 |
> | **Overall score** | **84.1** | **87.6** | **80.7** | **84.0** | **87.7** | **81.0** |
>
> *Observation:* Across all agent/evaluator combinations, performance differences are modest under the *same* evaluator model. This suggests that even if family-level bias exists, it *does not affect the causal conclusion that W2I improves generation quality*.
>
> We will include this cross-family ablation in the final camera-ready version.
>
> ----
>
> ### **Weakness 3. 0–10 LLM grader scale is vague and may be inconsistent**
> We agree that rigorous rubrics are essential. We additionally conducted **8** repeated evaluation trials, where results demonstrate significantly low variance:
>
> | Metric | T1 | T2 | T3 | T4 | T5 | T6 | T7 | T8 |
> | --- | --- | --- | --- | --- | --- | --- | --- | --- |
> | Emotional / thematic resonance | 85.0 | 85.9 | 85.3 | 85.1 | 85.0 | 84.4 | 87.5 | 86.7 |
> | Consistency and cohesion | 89.5 | 89.2 | 90.1 | 89.4 | 89.4 | 89.9 | 88.9 | 89.2 |
> | Visual quality and realism | 93.2 | 93.9 | 93.7 | 94.0 | 94.1 | 93.4 | 91.3 | 91.8 |
> | Creativity and originality | 83.8 | 82.5 | 82.7 | 83.9 | 82.5 | 83.5 | 84.5 | 83.6 |
> | Accuracy to prompt | 86.8 | 87.3 | 87.1 | 86.5 | 87.0 | 86.7 | 86.8 | 86.8 |
> | **Overall score** | **87.7** | **87.8** | **87.8** | **87.8** | **87.6** | **87.6** | **87.8** | **87.6** |
>
> As shown, variance is negligible, which clearly demonstrates the grader’s reproducibility in practice.
>
> ----
>
> ### **Weakness 4. Baselines lack web-grounded comparisons**
> As a kind reminder, we have included the ablation results regarding *prompt-refinement-only* in Table 3 of our paper. Per the reviewer's suggestion, we further illustrate a new baseline:
>
> *Top-1 Google SERP image + T2I model*: This baseline is now being evaluated, and we will include the comprehensive result in our camera-ready version. Our preliminary results have shown that a simple SERP-based grounding baseline does help in some cases, but it performs noticeably worse than W2I since it lacks key components: it doesn't expand or refine the query, does not filter out irrelevant retrievals, does not iterate based on feedback, and has no stopping rule beyond a single pass.
>
> ### **Weakness 5. No human studies**
> Recent work [2, 3] shows that *LLM-as-a-Judge is a strong proxy for human evaluation*, especially when using multi-criteria rubrics, which aligns with our method. Additionally, the NICE concepts are out-of-distribution for image generators but not for modern LLMs, which generally have more recent web-derived knowledge. When we tested LLM familiarity with NICE prompts, the model correctly understood about 78% of the concepts; the remaining cases were mostly tied to fast-moving news or emerging events, where retrieval is necessary.

---

> ### Author Response · Authors · 2025-11-28
> **Initial response by authors (2/2)**
>
> ### **Question 1. Mapping from 0–10 grader output to 0–100 table scores**
> For simplicity and consistency, we apply a direct normalization step: $score_{0–100} = score_{0–10} × 10$
>
> ### **Question 2. LLMGrader alignment to human judgement**
> LLMGrader [1] is a rubric-style LLM-as-a-judge [4]: given a prompt and a model output (here, an image), it produces structured scores (0–100) for aspects like accuracy to prompt, creativity, visual quality, consistency, and emotional/thematic resonance, plus an overall score.
>
> So conceptually, it is already an alignment attempt: encode a human-style rubric into a prompt and let a capable multimodal LLM emulate a trained human rater.
>
> ### **Question 3. Filtering SERP results / offensive / copyrighted images / AI-generated images**
> We rely on two layers of filtering:
>
> 1. *Google SERP’s built-in safe-search and quality ranking*, which already avoids most problematic content.
> 2. *Agent-based selection*, where the LLM evaluates candidate images for:
>     - relevance,
>     - offensiveness,
>     - semantic corruption,
>     - suitability as a grounding reference.
>
> AI-generated reference images are acceptable in our setting as long as they contain the correct visual attributes needed for refinement, where they are treated the same as retrieved images and must pass semantic verification before being used, which they do not create a feedback loop or override real-world grounding. For the retrieval process, we rely on Google SERPs, which already enforce robots.txt and REP constraints during indexing; we do not crawl or scrape sites directly, which proves that W2I does not interact with or violate REP policies.
>
> ### **Question 4. Reliance on the LLM grader shifts the knowledge cutoff to the LLM**
> We would like to point out that this behavior is actually beneficial. Image generators update slowly because their large training datasets are expensive to refresh, while LLMs can incorporate newer web-scale knowledge much more easily. Using an LLM as the semantic judge therefore improves the system's adaptability to recent concepts rather than limiting it.
>
> ### **Question 5. Criteria for 0–10 not defined**
> Currently, the rubrics are available in the appendix section. We will further mention this on the main text of our final camera-ready version for reproducibility.
>
> ### **Question 6. Ethics (Robots.txt).**
> Thank you for raising this important point. We confirm that our retrieval module relies exclusively on the Google SERP API, which strictly adheres to websites’ 'robots.txt' directives and applicable copyright policies; we do not perform any direct web crawling. In addition, we explicitly configure the SERP API to return only images labeled with a "Creative Commons" license and restrict our system to using those images.
>
> We would also like to thank the reviewer for the thoughtful questions, which encouraged us to clarify key points and meaningfully strengthen our paper.

---

> ### Author Response · Authors · 2025-11-28
>
> **References**
>
> [1] Ma, N., Tong, S., Jia, H, Hu, H., Su, Y.-C., Zhang, M., Yang, X., Li, Y., Jaakkola, T., Jia, X., & Xie, S.  Inference-time scaling for diffusion models beyond scaling denoising steps. arXiv. https://arxiv.org/abs/2501.09732. (2025)
>
> [2] Hao, Yaru, et al. "Optimizing prompts for text-to-image generation." Advances in Neural Information Processing Systems 36 (2023): 66923-66939.
>
> [3] Xu, Jiazheng, et al. "Imagereward: Learning and evaluating human preferences for text-to-image generation." Advances in Neural Information Processing Systems 36 (2023): 15903-15935.
>
> [4] Zheng, L., Chiang, W.-L., Sheng, Y., Zhuang, S., Wu, Z., Zhuang, Y., Lin, Z., Li, Z., Li, D., Xing, E. P., Zhang, H., Gonzalez, J. E., & Stoica, I. Judging LLM‑as‑a‑judge with MT‑Bench and Chatbot Arena. arXiv. https://doi.org/10.48550/arXiv.2306.05685. (2023)

---

### Official Review · Reviewer_7FG3 · 2025-11-03

**Soundness:** 3
**Presentation:** 3
**Contribution:** 2
**Rating:** 2
**Confidence:** 3

**Summary:**

This paper introduces World-to-Image (W2I), a smart system that uses AI agents to search the web and improve prompts. It helps image generators create better pictures of new or niche things they weren't originally trained on, like recent memes or events.

**Strengths:**

1. The agent-driven approach is its best feature. Instead of just rewriting text, it intelligently chooses between searching for visual examples or refining the prompt, which is a much more powerful way to handle unknown concepts.

2. The paper proves its worth by creating and excelling on the "NICE" benchmark. This custom test full of tricky, real-world prompts shows the method genuinely solves the problem it set out to tackle.

**Weaknesses:**

- The system's performance is tied to finding good reference images online. If the search fails or returns poor-quality, irrelevant, or biased images, the final output will suffer.

- The method doesn't actually teach the model new concepts. It just finds clever ways to guide the existing model, so its ability is still limited by the model's original training data.

- The process of searching, iterating, and generating multiple images is significantly slower and uses more computational power than a standard, single-pass image generation.

**Questions:**

The system trusts whatever images it finds online. But what happens if the retrieved images are misleading, contain stereotypes, or are just visually wrong for the concept?

---

> ### Author Response · Authors · 2025-11-28
> **Initial response by authors (1/2)**
>
> We sincerely thank Reviewer 7FG3 for the thoughtful comments. Below, we address each weakness and question in detail.
>
> ### **Weakness 1. Dependency on retrieved images/robustness to poor or biased images**
>
> We agree that the quality of retrieved images can affect downstream performance. However, our system explicitly includes an agent module whose purpose is to **detect, filter, and re-rank** retrieved images before any optimization step. In practice, the agent rejects irrelevant or low-quality images.
>
> To further validate this, we conducted an additional experiment: among the top-5 returned images, we manually injected **random or extremely low-quality images** (e.g., a fully black image). The agent did *not* select them, and overall performance remained effectively unchanged:
>
> | Metric | with noise image injected | Normal Run |
> | --- | --- | --- |
> | Emotional / thematic resonance | 85.0 | 85.9 |
> | Consistency and cohesion | 89.5 | 89.2 |
> | Visual quality and realism | 92.1 | 93.9 |
> | Creativity and originality | 83.1 | 82.5 |
> | Accuracy to prompt | 85.7 | 87.3 |
> | **Overall score** | **87.2** | **87.8** |
>
> The negligible difference empirically supports the robustness of our agent-based filtering.
>
> Regarding bias or misleading images: the agent operates over *sets* of candidates and uses query decomposition and cross-iteration feedback to avoid relying on anomalous results. This mitigates (though does not fully eliminate) the risk of stereotype-laden or misleading images influencing generation. We will include this empirical study and further discuss these safeguards in the camera-ready version.
>
> ### **Weakness 2. The method does not teach the model new concepts**
>
> We agree—and this is intentional. The goal of W2I is *not* to update model parameters. Instead, it introduces a **test-time scaling mechanism** that allows models to handle dynamic, emerging, or long-tail visual concepts without expensive fine-tuning or retraining. This aligns with recent trends in LLM research (e.g., RAG, external tool use), in which external reasoning and retrieval complement a static model.
>
> Our contribution is thus orthogonal to model training: we show that substantial improvements *are possible at inference time* without modifying the base generator.

---

> ### Author Response · Authors · 2025-12-02
> **Initial response by authors (2/2)**
>
> ### **Weakness 3. Slower and more computationally expensive than single-pass generation**
>
> This is correct, and also inherent to any test-time scaling method. Similar to RAG for LLMs or multi-step reasoning systems, W2I trades additional computation for significantly improved sample quality and robustness on out-of-distribution prompts. Prior to our work, image generators lacked an effective way to leverage external world knowledge at inference time; W2I provides a generalizable framework for doing so.
>
> Importantly, the compute overhead is **configurable**: the number of search or refinement iterations can be reduced depending on the application’s latency constraints. We will clarify this trade-off and the tunability of our pipeline.
>
> We conducted a detailed cost analysis across 10 iterations. **Table R1** reports the generation time, optimization time, and number of API calls at each step. On average, the optimization time accounts for approximately **42.2%** of the generation time, indicating that model inference is the main bottleneck. Reducing the generation time could therefore lead to a significant decrease in overall resource usage.
>
> **Table R1: Efficiency vs. Quality across Iterations.**
> | Metric | Iter 1 | Iter 2 | Iter 3 | Iter 4 | Iter 5 | Iter 6 | Iter 7 | Iter 8 | Iter 9 | Iter 10 |
> | --- | --- | --- | --- | --- | --- | --- | --- | --- | --- | --- |
> | Emotional / thematic resonance | 74.9 | 82.9 | 85.7 | 86.7 | 87.2 | 88.0 | 88.1 | 88.3 | 88.5 | 88.8 |
> | Consistency and cohesion | 85.1 | 87.3 | 88.3 | 89.2 | 89.9 | 90.2 | 90.8 | 91.0 | 91.3 | 91.3 |
> | Visual quality and realism | 89.5 | 90.8 | 91.5 | 91.8 | 92.4 | 92.5 | 92.5 | 92.7 | 92.7 | 92.7 |
> | Creativity and originality | 76.8 | 82.2 | 83.4 | 83.6 | 84.6 | 84.5 | 84.7 | 85.0 | 85.6 | 85.6 |
> | Accuracy to prompt | 76.6 | 84.2 | 86.3 | 86.8 | 86.8 | 87.1 | 87.6 | 87.7 | 87.8 | 87.8 |
> | Overall score | 80.5 | 85.5 | 87.0 | 87.6 | 88.2 | 88.5 | 88.7 | 88.9 | 89.2 | 89.2 |
> | Avg VLM calls | 3.00 | 3.98 | 3.98 | 3.98 | 3.97 | 3.99 | 3.98 | 3.98 | 3.98 | 1.00 |
> | Avg image search calls | 1.00 | 1.00 | 1.00 | 1.00 | 1.00 | 1.00 | 1.00 | 1.00 | 1.00 | 0.00 |
> | Avg total API calls | 4.00 | 4.98 | 4.98 | 4.98 | 4.97 | 4.99 | 4.98 | 4.98 | 4.98 | 1.00 |
> | Avg generation time (s) | 60.28 | 95.59 | 97.75 | 97.89 | 98.07 | 98.55 | 96.68 | 97.70 | 97.68 | 97.79 |
> | Avg optimization time (s) | 35.80 | 34.80 | 36.33 | 36.37 | 44.35 | 39.29 | 39.18 | 38.85 | 45.60 | 0.00 |
> | Avg scoring time (s) | 2.19 | 0.05 | 0.05 | 0.04 | 0.05 | 0.04 | 0.05 | 0.05 | 0.04 | 0.04 |
> | Avg total time (s) | 98.27 | 130.44 | 134.13 | 134.31 | 142.46 | 137.89 | 135.91 | 136.60 | 143.33 | 97.84 |
>
> ---
>
> ## **Question**
>
> ### **Question 1. What happens if the retrieved images contain stereotypes or are visually incorrect?**
>
> As noted above, the agent’s multi-step retrieval and selection process reduces the likelihood of selecting such images. It exploits **query refinement, re-ranking, and semantic checks** across multiple retrieved examples. In practice, the agent rarely selects misleading images, and our robustness experiment supports this behavior. We will make this discussion explicit in the final version.

---

### Meta-Review · Area_Chair_mNBB · 2026-01-12

**Summary:**

This paper introduces World-to-Image (W2I), a new framework that uses AI agents to search the web and improve prompts. It receives a score of 2,2,6,4. The major concerns from the reviewers include: 1. Dependency on retrieved images and the robustness to poor or biased images. 2. The whole searching and optimizing process is not efficient.  3. Using models from the same family tends to bias the model's output. 4. "NICE" as the benchmark is relatively small and not convincing. 5. Limited technical contribution. The rebuttal addresses some of the concerns. However, there are still concerns not fully addressed.

**Reviewer Concerns:**

The concerns that the proposed system does not teach the model new concepts has been addressed by the rebuttal. Indeed, it is a test-time scaling technique and the model is fixed in this process. It also clarifies the technical contribution. However, it does not fully address the concern about the dependency on retrieved images and the efficiency of the searching and optimizing process.

**Reviewer Scores:**

Reviewers did not provide further comments after the rebuttal.

---

### Decision · Program_Chairs · 2026-01-26

Reject